# MCTP is an ER-resident calcium sensor that stabilizes synaptic transmission and homeostatic plasticity

Özgür Genç[1], Dion K Dickman[1,2], Wenpei Ma[2], Amy Tong[1], Richard D Fetter[1], Graeme W Davis[1]*

[1]Department of Biochemistry and Biophysics, Kavli Institute for Fundamental Neuroscience, University of California, San Francisco, San Francisco, United States; [2]Department of Biological Sciences, University of Southern California, Los Angeles, United States

**Abstract** Presynaptic homeostatic plasticity (PHP) controls synaptic transmission in organisms from *Drosophila* to human and is hypothesized to be relevant to the cause of human disease. However, the underlying molecular mechanisms of PHP are just emerging and direct disease associations remain obscure. In a forward genetic screen for mutations that block PHP we identified *mctp* (Multiple C2 Domain Proteins with Two Transmembrane Regions). Here we show that MCTP localizes to the membranes of the endoplasmic reticulum (ER) that elaborate throughout the soma, dendrites, axon and presynaptic terminal. Then, we demonstrate that MCTP functions downstream of presynaptic calcium influx with separable activities to stabilize baseline transmission, short-term release dynamics and PHP. Notably, PHP specifically requires the calcium coordinating residues in each of the three C2 domains of MCTP. Thus, we propose MCTP as a novel, ER-localized calcium sensor and a source of calcium-dependent feedback for the homeostatic stabilization of neurotransmission.

*For correspondence: graeme.davis@ucsf.edu

## Introduction

Homeostasis can be defined as the ability of a cell, or system of cells, to detect a perturbation and, in the continued presence of that perturbation, generate a compensatory response that precisely restores baseline function. It is well established that homeostatic signaling stabilizes the active properties of excitable cells, including neurons and muscle (*Davis, 2006*, *2013*; *Davis and Müller, 2015*; *Marder, 2011*; *Turrigiano, 2011*). It is also generally accepted that neuronal homeostatic signaling will be essential for the life-long robustness of brain function and animal behavior (*Davis, 2006*, *2013*). Indeed, numerous reviews have speculated about potential connections between homeostatic plasticity and psychiatric or neurological disease, including autism spectrum disorders (*Mullins et al., 2016*; *Nelson and Valakh, 2015*), schizophrenia (*Dickman and Davis, 2009*) and epilepsy (*Noebels, 2015*). But, the cellular and molecular underpinnings of neuronal homeostasis are only just beginning to emerge through forward genetic and biochemical screening efforts (*Dickman and Davis, 2009*; *Davis and Müller, 2015*). Delineating these signaling systems and defining how they succeed or fail, in health and disease, will be essential before concrete connections between homeostatic plasticity and disease can be established.

Signaling mechanisms that are capable of homeostatic control are known to incorporate basic elements including sensors that can detect a perturbation, error signals that define the magnitude of a perturbation, feedback and feed-forward signaling loops, and mechanisms of signal integration (*Alon, 2007*; *Cowan et al., 2014*; *Mitrophanov and Groisman, 2008*; *Lemmon et al., 2016*). For

example, we have a detailed understanding of the homeostatic signaling systems that control blood glucose levels and organismal metabolism. Hundreds of proteins and enzymes are organized into intersecting regulatory pathways, inclusive of feedback and feed-forward signaling elements that are dependent upon reversible and irreversible enzymatic reactions. By comparison, the biological basis for the homeostatic control of neural function remains almost completely uncharacterized. It seems essential that new homeostatic plasticity genes be identified so that entire signaling pathways can be delineated and new regulatory mechanisms defined. Here, we identify and characterize a novel presynaptic calcium sensor for homeostatic synaptic plasticity. We present evidence that this calcium sensor resides on the membranes of the presynaptic endoplasmic reticulum (ER) and, as such, defines a new homeostatic signaling relationship between the ER and the presynaptic neurotransmitter release mechanism.

The homeostatic modulation of presynaptic neurotransmitter release, termed presynaptic homeostatic plasticity (PHP), is highly conserved from *Drosophila* to human (*Davis, 2013*; *Davis and Müller, 2015*). In response to a perturbation that disrupts the function of postsynaptic neurotransmitter receptors, there is a compensatory increase in presynaptic neurotransmitter release that precisely offsets the change in neurotransmitter receptor function and maintains postsynaptic excitation at baseline, set point levels (*Davis, 2013*; *Davis and Müller, 2015*). PHP can be rapidly induced on a time scale of seconds to minutes, can precisely adjust the presynaptic release mechanism to offset a wide range of postsynaptic perturbations and can be stably maintained for the life of an organism; months in *Drosophila* (*Mahoney et al., 2014*) and decades in human (*Cull-Candy et al., 1980*).

It remains a formidable challenge to define signaling mechanisms that are capable of the rapid, accurate and persistent modification of presynaptic neurotransmitter release observed during PHP (*Davis, 2006*, *2013*). Recent advances include the identification and involvement of a presynaptic ENaC channel capable of mediating analogue modulation of presynaptic calcium influx (*Younger et al., 2013*), the RIM/RIM-Binding Protein scaffold (*Müller et al., 2012*; *Müller et al., 2015*) and a novel presynaptic innate immune receptor, PGRP (*Harris et al., 2015*). Although these genes are causally linked to the induction and expression of PHP, the adaptive controls that are able to modulate the vesicle release mechanism in an accurate and persistent manner remain poorly defined. For example, how does a presynaptic scaffolding protein such as RIM participate in a regulated, rheostat-like control of presynaptic release? Homeostatic signaling systems are often built upon feedback signaling. But, to date, the molecular basis of homeostatic feedback within the presynaptic terminal remains a mystery.

In the original forward genetic screen for candidate PHP genes (*Dickman and Davis, 2009*) we identified mutations that disrupted a gene encoding a C2 domain containing protein of unknown function, termed MCTP (Multiple C2 Domain Proteins with Two Transmembrane Regions). MCTPs are evolutionarily conserved proteins with three C2 domains and two C-terminal membrane-spanning domains (*Shin et al., 2005*). The C2 domains are unusual in that they are able to bind calcium with relatively high affinity (1–2.5 µM) and do so in the apparent absence of phospholipid binding (*Shin et al., 2005*). In *C. elegans*, genetic deletion of *mctp* is embryonic lethal (*Maeda et al., 2001*). In mammals, *mctp* is expressed in the brain and spinal cord and genetic mutations in *mctp* have been linked to bipolar disorder (*Djurovic et al., 2009*; *Scott et al., 2009*). Beyond this, little is known about the function of MCTP. Here, we demonstrate that *Drosophila* MCTP is resident on the membranes of the endoplasmic reticulum that extends from the soma to the presynaptic terminal. We provide evidence that MCTP functions in motoneurons as a potential source of calcium-dependent feedback control that sustains both baseline neurotransmitter release and presynaptic homeostatic plasticity.

## Results

### Identification of *mctp* as a homeostatic plasticity gene

A large-scale forward genetic screen is being employed to identify genes that are required for PHP (*Dickman and Davis, 2009*; *Younger et al., 2013*; *Müller et al., 2015*). This screen is based on the application of the glutamate receptor antagonist philanthotoxin-433 (PhTx) to the NMJ at sub-blocking concentrations (4–20 µM). Following application of PhTx, miniature excitatory postsynaptic potential amplitudes (mEPSP) are initially decreased by ~50% and there is a parallel decrease in the

amplitude of evoked excitatory postsynaptic potentials (EPSP) (*Frank et al., 2006*). During the next 10 min, in the continued presence of PhTx, mEPSP amplitudes remain inhibited while EPSP amplitude gradually increases, ultimately reaching amplitudes observed prior to the application of PhTx (*Frank et al., 2006*). Thus, enhanced presynaptic release rapidly and precisely offsets the magnitude of postsynaptic glutamate receptor inhibition, homeostatically restoring EPSP amplitudes. This process has been observed at mammalian central and peripheral synapses and is conserved at the human NMJ (*Davis and Müller, 2015* for review). Recently, the rapid induction and rapid reversal of PHP were also documented at the mouse NMJ (*Wang et al., 2016*). To identify the molecular mechanisms of PHP, we have screened more than 2000 individual gene mutations for impaired PHP using electrophysiological recordings of synaptic transmission as a primary assay (*Dickman and Davis, 2009*; *Müller et al., 2012*; *Younger et al., 2013*). To date, very few genes have been identified that are required for both the rapid induction and sustained expression of PHP. This screen identified a transposon insertion in the *mctp* gene that blocks the rapid induction of PHP.

## MCTP is necessary for both PHP and baseline presynaptic release

The $mctp^1$ transposon insertion resides in the non-coding, 3' untranslated region of the *mctp* gene and is predicted to be a loss of function allele. We acquired a second transposon insertion ($mctp^2$) that resides within a coding exon that is predicted to be a molecular null mutation as well as a deficiency chromosome that uncovers the *mctp* locus (*Figure 1A*). In addition, using CRISPR-mediated gene editing, we generated a single nucleotide insertion that creates a frame shift and an early stop codon at amino acid 132, an *mctp* mutation that is predicted to be a molecular null ($mctp^{OG9}$). All three mutations are viable to the adult stage, allowing recordings to be made from normally sized NMJ at the third instar larval stage.

First, we demonstrate that PHP is completely blocked in all three *mctp* mutants, as well as when $mctp^1$ is placed in trans to the *mctp* deficiency (*Figure 1B–C*). More specifically, application of sub-blocking concentrations of PhTx causes a significant decrease in mEPSP amplitude in wild type and in all of the *mctp* mutant combinations that we tested (*Figure 1D*; Student's t-test; $p<0.01$ for each genotype, comparing the absence versus presence of PhTx). In wild type, there is a homeostatic increase in quantal content (EPSP/mEPSP; *Figure 1F*; $p<0.01$) that precisely offsets the decrease in mEPSP amplitude and restores EPSP amplitudes to baseline (*Figure 1B,E*). However, in all four *mctp* mutant combinations, the homeostatic increase in quantal content is blocked (*Figure 1F*; $p>0.05$ for each genotype relative to baseline in the absence of PhTx) and EPSP amplitudes remain smaller than baseline (*Figure 1E*). Thus, *mctp* is essential for the rapid induction of PHP.

An analysis of baseline transmission in response to single action potential stimulation, under these recording conditions (0.3 mM $[Ca^{2+}]_e$), reveals a significant deficit in presynaptic release in all of the *mctp* mutant combinations that we tested. There is no difference in mEPSP amplitude in any genotype (*Figure 1D*, solid bars; $p>0.05$). However, all mutant alleles have diminished average EPSP amplitudes compared to wild type (*Figure 1E*, solid bars; $p<0.05$). Calculated quantal content is significantly decreased in $mctp^1$, $mctp^2$ and $mctp^{OG9}$ (*Figure 1F*, solid bars; $p<0.05$). An additional analysis of baseline neurotransmission across a 5-fold range of extracellular calcium concentration, performed under two-electrode voltage clamp configuration and inclusive of an analysis of short-term synaptic modulation, is shown below. Importantly, while there are relatively mild effects on baseline transmission under these conditions, the expression of PHP is completely blocked.

## MCTP functions in motoneurons during PHP

To confirm that MCTP is necessary for PHP and to define where MCTP functions, pre- versus post-synaptically, we performed genetic rescue experiments. A full-length *mctp* transgene was generated and expressed under *UAS* control using either the neuron-specific *Gal4* driver *elav-Gal4* or the muscle specific driver *MHC-Gal4*. Neuron-specific expression of *UAS-mctp* in the $mctp^1$ mutant background ('rescue neuron' in *Figure 1G–J*) fully rescues PHP as demonstrated by a large increase in quantal content and the restoration of EPSP amplitudes to baseline values in the presence of PhTx (*Figure 1G–J*; $p>0.5$ compared to wild type for mEPSP, EPSP and quantal content values). By contrast, muscle-specific expression of *UAS-mctp* ('rescue muscle' in *Figure 1G–J*) does not rescue PHP. Indeed, we note that while expression of *UAS-mctp* in muscle alone has no effect on quantal content compared to wild type ($p>0.1$), in the presence of PhTx the quantal content is reduced below levels

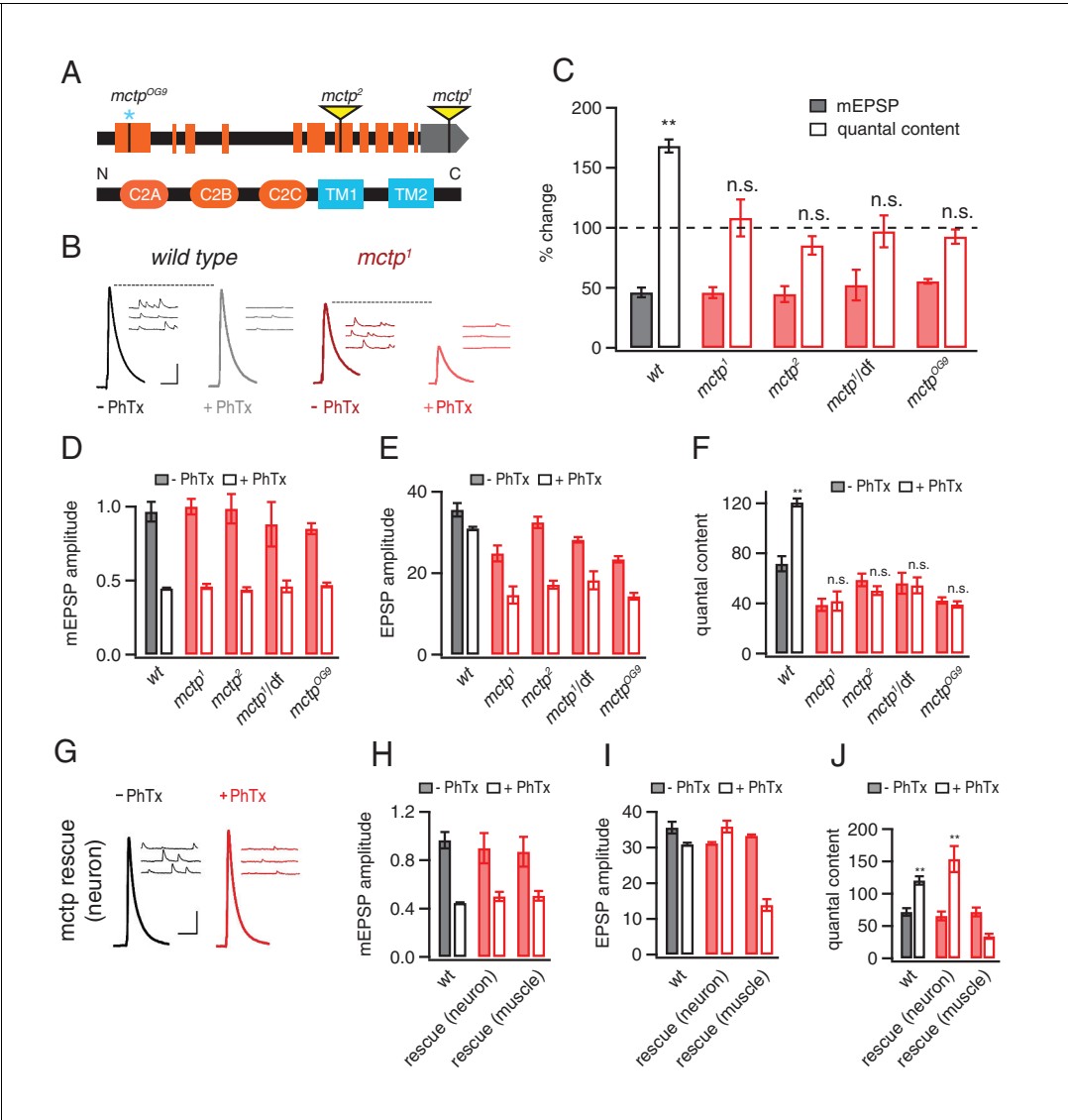

**Figure 1.** MCTP is a multiple C2 domain protein, which is necessary for the synaptic homeostasis. (A) Top: schematic of the *Drosophila mctp* locus. The position of transposon insertion mutations (yellow triangles) and CRISPR-induced mutation (blue asterisk) are shown. Bottom: Diagram of the MCTP protein, which contains three C2 domains (orange round rectangles) and two transmembrane domains (blue rectangles). (B) Representative traces showing EPSPs and mEPSPs from *mctp and wild type* in the absence (left, black or dark red) or presence (light gray or light red) of philanthotoxin (PhTx) as indicated below. Scale bar: 4 mV/7 mV; 25 ms/60 ms. (C) Average percent change in mEPSP amplitude (filled bars) and quantal content (open bars) for the indicated genotypes, calculated as the percent change of each genotype in the presence of PhTx compared to baseline in the absence of PhTx. *wild type* (*wt*): n = 10 (-PhTx), n = 10 (+PhTx); *mctp[1]*: n = 11 (-PhTx), n = 8 (+PhTx); *mctp[2]*: n = 8 (-PhTx), n = 9 (+PhTx); *mctp[1]/df*: n = 8 (-PhTx), n = 7 (+PhTx); *mctp[OG9]*: n = 20 (-PhTx), n = 37 (+PhTx). Statistical comparisons are made comparing values in the presence or absence of PhTx within each genotype (Student's t-test). (D–F) Quantification of mEPSP amplitude (D), EPSP amplitude (E) and quantal content (F) in the absence and presence of PhTx (as indicated). Statistical comparisons as in (C). (G) Representative traces for EPSPs and mEPSPs showing the restoration of the presynaptic homeostasis upon expression of a *UAS-mctp* transgene in neurons (*elav-Gal4*; black baseline; red +PhTx) Scale bar: 4 mV/7 mV; 25 ms/60 ms. (H–J) Quantification of average mEPSP amplitude, EPSP and quantal content for wild type (n = 10 without PhTx; n = 10 + PhTx), neural rescue (*elav-Gal4*) (n = 13 without PhTx; n = 10 + PhTx) and muscle rescue (*MHC-Gal4*) of *mctp[1]* mutant (n = 11 without PhTx; n = 9 + PhTx) in the absence and presence of philanthotoxin (filled and open bars, respectively). Student's t-test **p<0.01, n.s. p>0.05.

observed at baseline in the absence of PhTx, suggesting a dominant interfering effect of postsynaptic expression that is somehow specific to the condition of PhTx application. Regardless, taken together, these data confirm that *mctp* is essential for the rapid induction of PHP and that MCTP activity is necessary in the presynaptic neuron for PHP.

## MCTP is essential for the long-term maintenance of PHP

Mutations in the muscle-specific glutamate receptor subunit *GluRIIA* decrease mEPSP amplitudes and induce a homeostatic potentiation of presynaptic release (*Petersen et al., 1997*). This manipulation has been interpreted to reflect the long-term maintenance of PHP and represents an independent experimental method to induce PHP. Here, we confirm that the *GluRIIA* mutant has decreased mEPSP amplitudes and a robust, homeostatic increase in quantal content (p<0.01). However, when PHP is assessed in the *mctp¹,GluRIIA* double mutant, PHP is completely blocked (*Figure 2*). More specifically, in *mctp¹,GluRIIA* double mutants, mEPSP amplitudes are decreased to levels identical to that observed in the *GluRIIA* mutant (*Figure 2*). But, unlike the *GluRIIA* mutant, there is a failure to increase quantal content (*Figure 2*). The specificity of this effect is confirmed by genetic rescue experiments. Presynaptic expression of *UAS-mctp* fully restores PHP in the *mctp¹,GluRIIA* double mutant background (*Figure 2*). We note that quantal and EPSP amplitudes in the *mctp¹,GluRIIA* double mutants are diminished below what is predicted by a simple block of PHP (*Figure 2B,D*). Since both EPSP amplitude and quantal content are fully restored by presynaptic expression of *UAS-mctp* in the *mctp¹,GluRIIA* double mutant, this effect can be attributed solely to the loss of *mctp*. This effect could reflect an additional activity of MCTP that is required for the consolidation or maintenance of PHP. This possibility will be the focus of future studies.

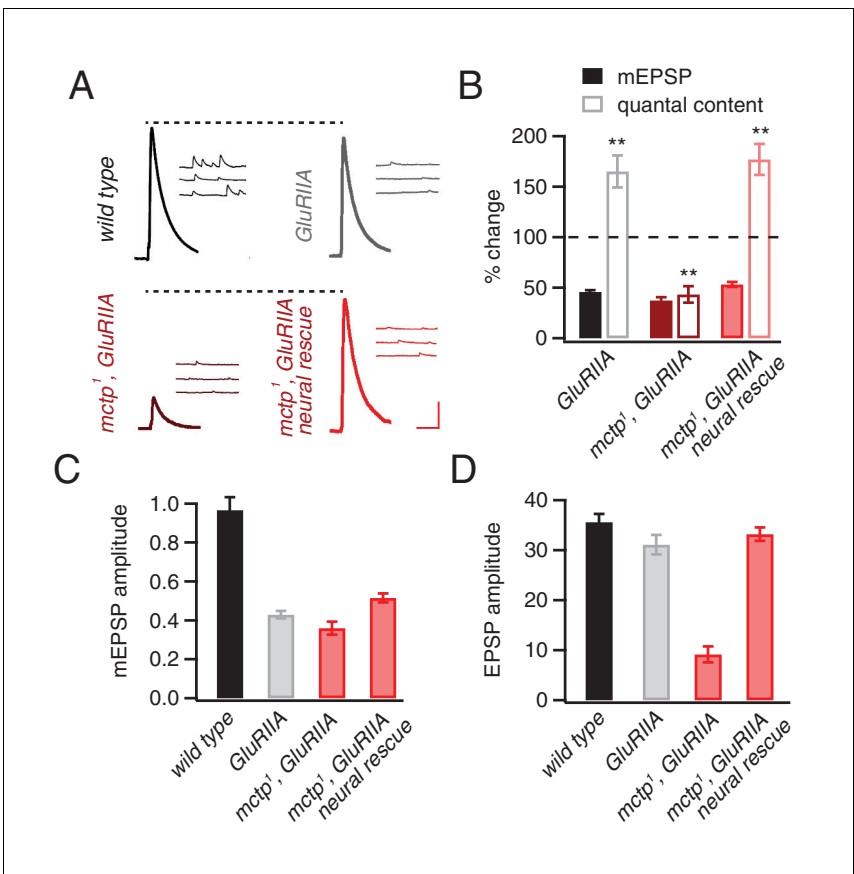

**Figure 2.** *mctp* is necessary for the sustained expression of homeostasis. (**A**) Representative traces of EPSP and mEPSP are shown for the following genotypes: wild-type (left, top), *GluRIIA* mutant (right, top), *mctp¹,GluRIIA* double mutant (left, bottom) and neural rescue of *mctp¹,GluRIIA* double mutant with *UAS-mctp* driven by *elav-Gal4* (right, bottom). Scale bar: 4 mV/7 mV; 25 ms/60 ms. (**B**) Average percent changes in mEPSP amplitude (filled bars) and quantal content (open bars) for indicated genotypes compared to *wild type* (wild type: n = 10; *GluRIIA*: n = 8; *mctp¹,GluRIIA*: n = 7; *mctp¹,GluRIIA* neural rescue: n = 10). (**C–D**) Average mEPSP amplitudes and quantal content are shown for genotypes as in (**A**). Student's t-test ** for p<0.01, n.s. for p>0.05.

## MCTP is enriched on ER membranes within the soma, dendrites, axon and presynaptic nerve terminal

To examine MCTP protein localization, we visualized an epitope tagged MCTP protein (MCTP-Flag; see Materials and methods). Efforts to generate MCTP-specific antibodies were unsuccessful. We note that *UAS-mctp* expression fully rescues the mutant phenotype. MCTP has two highly conserved transmembrane segments, implying localization to a cellular membrane system. We now demonstrate that MCTP-Flag localizes to the internal membranes of the endoplasmic reticulum (*Figure 3*). We expressed *UAS-mctp-flag* in a pair of identified motoneurons that are segmentally repeated in the larval central nervous system using the *MN1-Gal4* driver (see Materials and methods). Within the soma, MCTP concentrates within a reticulated, peri-nuclear membrane system (*Figure 3A*) and co-localizes with co-expressed ER-resident KDEL-RFP (not shown), indicative of ER localization. MCTP-Flag is observed to extend into the axon on similarly reticulated internal membranes (*Figure 3A* and inset). Co-staining with anti-HRP to label the neuronal plasma membrane confirms that these are a membrane system within the axonal plasma membrane (not shown). MCTP-Flag extends throughout the dendrites (*Figure 3A*) and then extends all the way to the presynaptic nerve terminal (*Figure 3B*). This is entirely consistent with recent studies in *Drosophila* and mammalian systems showing that the smooth ER extends throughout the soma, dendrite, axon and presynaptic terminal (*de Juan-Sanz et al., 2017*; *Summerville et al., 2016*). Consistent with this prior work, we demonstrate co-localization of MCTP-Flag with the ER resident protein HDEL-GFP (*Summerville et al., 2016*) in presynaptic terminals (*Figure 3B*, see also inset). In *Figure 3B* (inset) we show near perfect co-localization of MCTP-Flag with ER tubules identified by HDEL-GFP. In this experiment, the HDEL-GFP was imaged live to ensure proper ER morphology. The preparation was lightly fixed and stained for MCTP-Flag, showing precise co-localization of the markers and good preservation of the general ER morphology under our fixation conditions. In axons, MCTP-Flag also co-localizes with HDEL-GFP (*Figure 3D*).

To further investigate the sequences that localize MCTP to the ER, we resorted to a structure function analysis of the MCTP transmembrane domains, using heterologous insect cells (S2 cells). We demonstrate that the isolated transmembrane domains of MCTP are sufficient to localize GFP to the ER, defined by co-localization with ER-tracker (*Figure 3E*, compare full length MCTP-GFP to TM only GFP). Each image in *Figure 3E* is a single confocal section through the cell, highlighting the peri-nuclear organization of the ER. We then performed the converse experiment. When the two transmembrane domains of MCTP are removed from MCTP-GFP, the truncated protein no longer localizes to the ER and is found broadly distributed (*Figure 3E*, MCTP noTM GFP). Finally, we note that full length MCTP-GFP protein was never observed to reside on the cell plasma membrane (*Figure 3E* and data not shown).

We then explored co-labeling with other synaptic markers at the nerve terminal. First, we co-stained the terminal with MCTP-Flag and the active zone marker anti-BRP and imaged the NMJ using super-resolution structured illumination microscopy (*Figure 3C*). Single sections reveal that MCTP-Flag does not co-localize with anti-Brp, but does come into close proximity at many, though not all, active zones (*Figure 3C*). Next, we perform co-labeling with the synaptic vesicle marker anti-synaptotagmin1 (Syt1; *Figure 4*). We note that MCTP is essentially absent from clusters of synaptic vesicles, identified by anti-Syt1 staining (*Figure 4B* arrows). These data are consistent with retention of MCTP-Flag in the ER since over-expressed protein does not appear to extend to post-golgi membranes of synaptic vesicles. Co-labeling with a membrane marker (anti-HRP) again demonstrates that MCTP-Flag labeled membranes are often closely opposed to the neuronal plasma membrane (*Figure 4* inset; See also *Figure 3*). This is intriguing given the recent demonstration that the Extended Synaptotagmins (eSyts), which are members of the same extended gene family as MCTP, localize to sites of ER-plasma membrane contact (*Giordano et al., 2013*).

We acknowledge that our evidence for ER localization is based on protein overexpression. Ideally, the subcellular localization of a protein would be determined by comparison of multiple assays, including visualization of a fluorescently tagged protein in living cells and immunofluorescent localization in fixed, permeabilized tissue (*Stadler et al., 2013*; *Schnell et al., 2012*; *Moore and Murphy, 2009*; *Huang et al., 2014*). Both techniques have well-established caveats. The incorporation of a fluorescent tag into a protein of interest can alter protein localization, irrespective of whether the tagged protein is overexpressed or regulated by endogenous gene expression (*Stadler et al., 2013*;

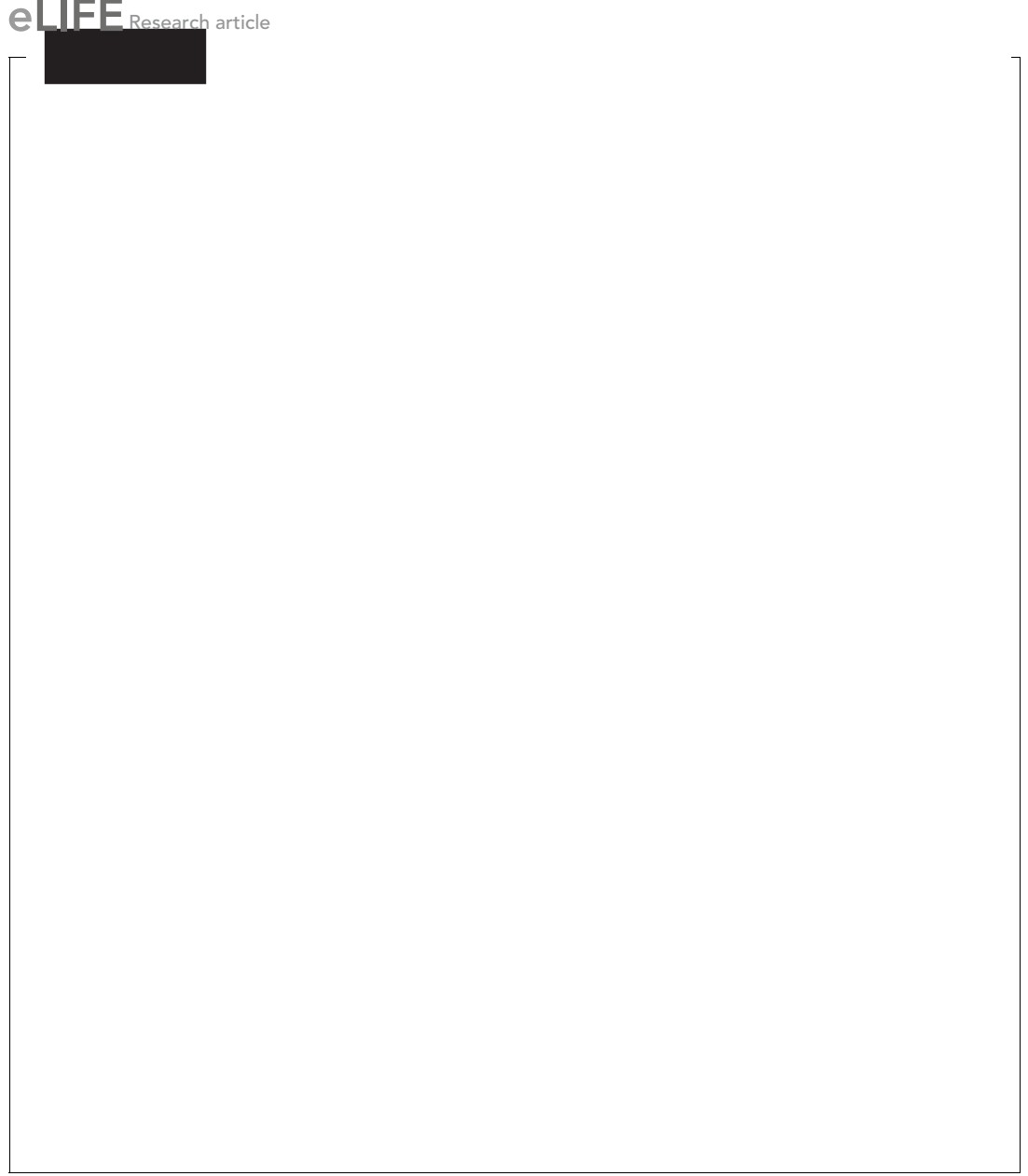

**Figure 3.** Epitope-tagged MCTP localizes to ER membranes. (**A**) Image shows the distribution of epitope-tagged MCTP in a single bilaterally symmetrical pair of motoneurons in the larval CNS (*MN1-Gal4>UAS-mctp-Flag*, green) The dotted line indicates the CNS midline and the numbers refer to inset images at right showing single confocal sections of peri-nuclear reticular membranes in the soma (1, 2) and the MCTP-labeled membranes that continue into the axon and dendrites (3). (**B**) Synaptic distribution of HDEL-GFP (green, top), imaged in a live preparation. Below, MCTP-Flag (red) is shown for the same NMJ following mild fixation and antibody labeling. The yellow box indicates the region shown at higher magnification to the right. The live HDEL-GFP and post-fixation MCTP-Flag identify the same tubular organization of the ER membranes within an individual synaptic bouton. No staining on the neuronal plasma membrane is evident. The NMJ is co-stained with a marker of the neuronal plasma membrane (anti-HRP, blue). Scale bar: 10 μm. (**C**) Structured illumination microscopy (SIM), single confocal section images of a synaptic bouton expressing the active zone marker Brp-GFP (driven by its endogenous promoter, green) and MCTP (*OK371-Gal4>UAS-MCTP-Flag*, red). Arrows indicate regions where MCTP resides in close proximity to Brp puncta, which define the T-bar structures that reside at the center of individual active zones. Scale bar: 1 μm. (**D**) Axon membranes from the peripheral nerve stained as in (**B**). Scale bar: 50 μm. (**E**) Single confocal sections of individual S2 cells transfected with MCTP-GFP and co-labeled with ER-tracker (red; see Materials and methods). At left, an S2 cell transfected with full length MCTP-GFP. At middle, an S2 cell transfected with a transgene expressing only the two-transmembrane region of MCTP linked to GFP (TM only-GFP). ER localization is retained. At right, an S2 cell transfected with a transgene expressing an MCTP-GFP transgene lacking the two-transmembrane domain and ER localization is lost (MCTP-noTM-GFP). Scale bar: 10 μm.

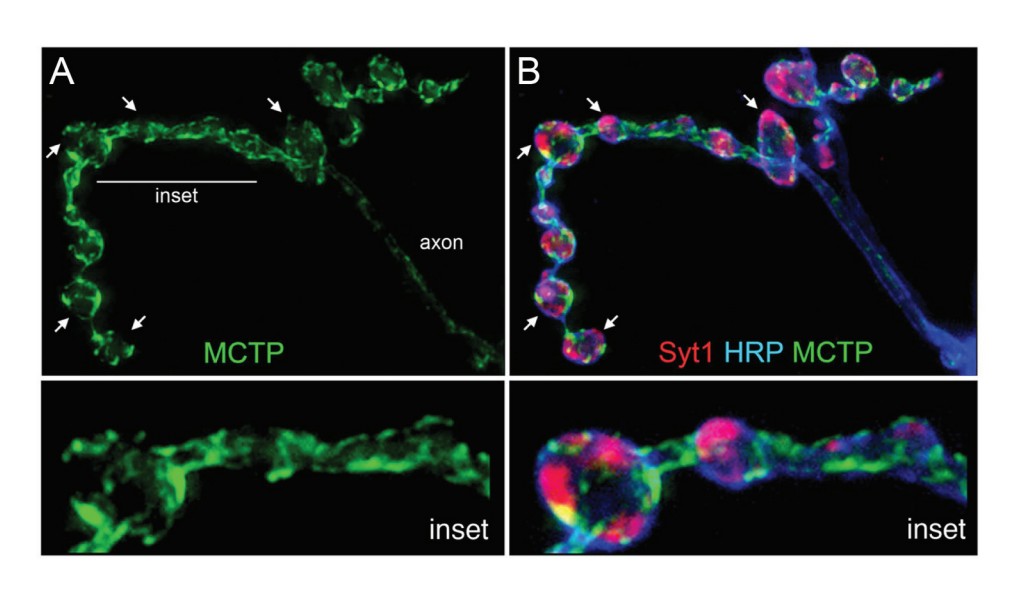

**Figure 4.** MCTP does not co-localize with the synaptic vesicle marker Syt1. (**A**) Synaptic distribution of MCTP-Flag (green) at the NMJ. (**B**) The synapse in (**A**) co-stained with markers of the plasma membrane (anti-HRP, blue) and synaptic vesicles (anti-Syt1, red). Arrows indicate regions where Syt1 immunoreactivity is abundant and MCTP-Flag is absent. Insets (below; 2.5x magnification) of the region indicated by a horizontal line (top left) reveal regions occupied by Syt1 staining that are devoid of MCTP-Flag at higher magnification. Additional analysis of NMJ bouton anatomy and cytoskeleton is presented in *Figure 4—figure supplement 1*.

The following figure supplement is available for figure 4:

**Figure supplement 1.** Normal NMJ Development in *mctp* mutants.

---

*Schnell et al., 2012*; *Moore and Murphy, 2009*; *Huang et al., 2014*). Similarly, fixation and tissue permeabilization can produce artifacts as can lack of antibody specificity (*Stadler et al., 2013*; *Schnell et al., 2012*; *Moore and Murphy, 2009*; *Huang et al., 2014*). We note, however, a recent study of more than 500 human proteins that documented ~80% concordance between fluorescent protein tagging and antibody labeling in human cell lines (*Stadler et al., 2013*). Thus, while acknowledging the caveats of each approach, there is evidence that each remains a reliable technique in modern cell biology. We were unable to verify MCTP localization with antibody staining. However, we do document identical MCTP localization in both neurons and S2 cells, rescue of the *mctp* mutant phenotype and we demonstrate the required function of the transmembrane region of MCTP for ER localization in S2 cells. Thus, based on these and related data, we conclude that MCTP is an ER resident protein, distributed on the ER throughout motoneurons.

## Loss of MCTP does not alter NMJ growth or morphology

Several ER-resident proteins have recently been implicated in the control of synaptic morphology (*Summerville et al., 2016*; *Moustaqim-Barrette et al., 2014*; *O'Sullivan et al., 2012*; *Lee et al., 2009*). Therefore, we quantified NMJ morphology in the *mctp* mutant background. We find a small, but statistically significant, increase in bouton number in both *mctp*[1] and *mctp*[2] (*Figure 4—figure supplement 1*). These changes are in the same direction as that observed for the *atlastin* and *spict* mutations, both ER resident proteins, but are substantially milder (*Summerville et al., 2016*; *Wang et al., 2007*). We also quantified active zone number, assessed by counting the number of anti-Brp puncta that identify the presynaptic T-bar, resident at individual active zones. There is also no significant change in active zone number or active zone density (*Figure 4—figure supplement 1*). Beyond this, the *mctp* mutant NMJ appears morphologically normal, without the appearance of satellite boutons (*Marie et al., 2004*) and without any evidence of NMJ degeneration (*Eaton et al.,*

*2002*). Previous studies have demonstrated that neuromuscular degeneration is associated with disruption of the presynaptic microtubule cytoskeleton, identified with anti-Futsch antibody staining (*Pielage et al., 2008*, *2011*). We find that anti-Futsch staining is qualitatively normal, consistent with normal NMJ growth and normal NMJ integrity (*Figure 4—figure supplement 1*). We conclude that impaired homeostatic plasticity in the *mctp* mutants is not a secondary consequence of impaired NMJ growth, stability or active zone number.

## MCTP is necessary for normal release probability and short-term release dynamics

Next, we investigated whether loss of MCTP alters baseline neurotransmission and short-term plasticity over a range of extracellular calcium concentrations. We repeated our analyses of baseline transmission using two-electrode voltage clamp to measure EPSC amplitudes across a 5-fold range of extracellular calcium (0.3–1.5 mM $[Ca^{2+}]_e$) and find that EPSC amplitudes are significantly impaired across this entire range (*Figure 5A*). Thus, in the $mctp^{OG9}$ mutant, there is an impaired calcium sensitivity of release (*Figure 5A*). This occurs without a change in the apparent calcium-cooperativity of release (see below), given that the magnitude of change is similar at all external calcium concentrations tested (*Figure 5A*). Consistent with the conclusion *mctp* alters baseline release, we also find a significant change in short-term release dynamics at 0.7, 1.0 and 1.5 mM $[Ca^{2+}]_e$ (*Figure 5B*). Indeed, at 1.0 mM $[Ca^{2+}]_e$ synaptic depression is observed at the wild type synapse compared to short-term facilitation at the *mctp* mutant synapse (*Figure 5B*). Similar alterations in short-term release

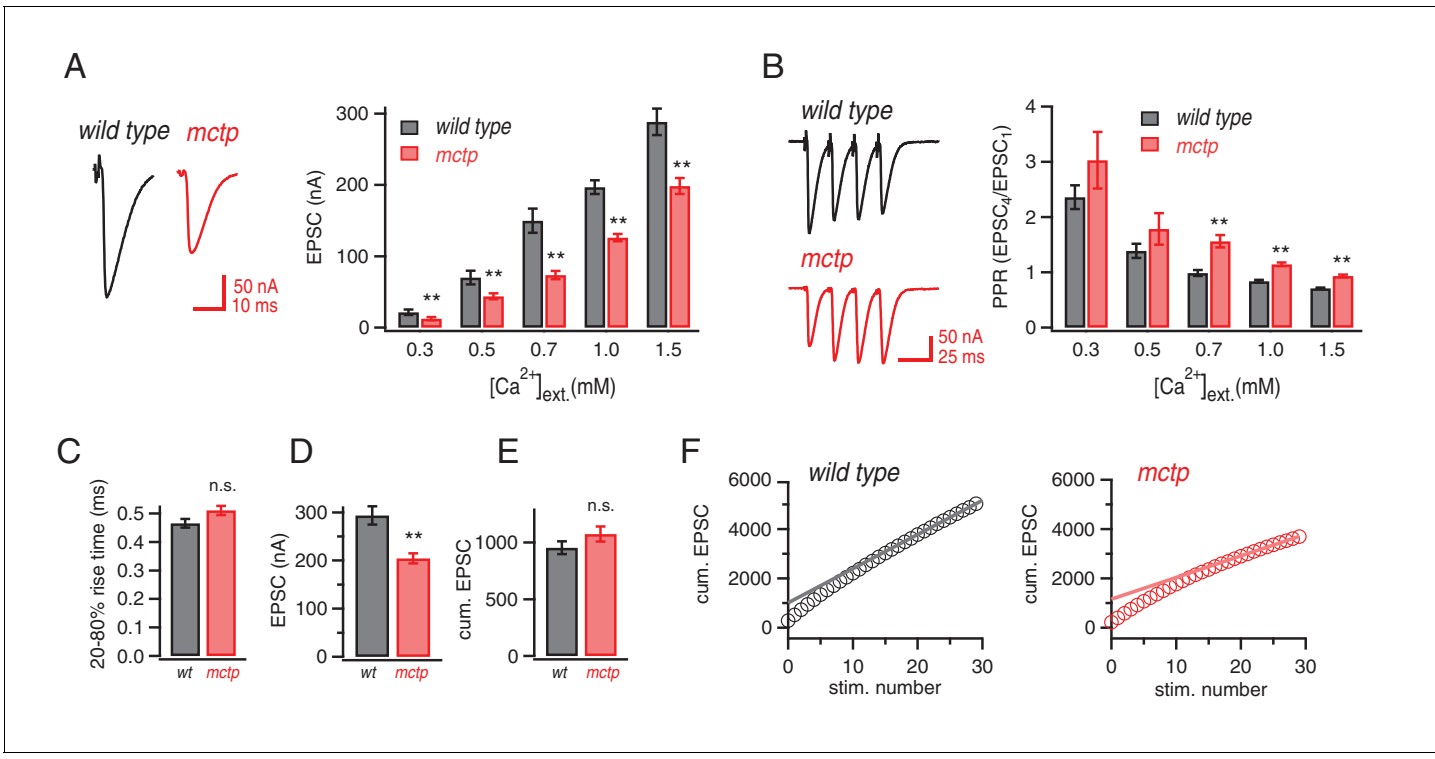

**Figure 5.** Altered baseline transmission and short-term plasticity in *mctp* mutants. (A) At left, representative EPSC traces for wild type (black) and $mctp^{OG9}$ (red). The plot shows average EPSC amplitudes for *wild type* (gray) and $mctp^{OG9}$ (red) at the indicated extracellular calcium concentrations. Sample sizes as follows: 0.3 mM: n = 10 (wt), n = 8 ($mctp^{OG9}$); 0.5 mM: n = 13 (wt), n = 11 ($mctp^{OG9}$); 0.7 mM: n = 7 (wt), n = 16 ($mctp^{OG9}$); 1.0 mM: n = 16 (wt), n = 13 ($mctp^{OG9}$); 1.5 mM: n = 12 (wt), n = 10 ($mctp^{OG9}$). The decrease in $mctp^{OG9}$ is significant at every calcium concentration (t-test). (B) At left, representative traces for short stimulus trains (60 Hz) for *wild type* (black) and $mctp^{OG9}$ (red). Scale bar: 50nA, 50 ms. The plot show average EPSC ratio (4thEPSC/1stEPSC) at the indicated calcium concentrations for *wild type* (gray) and $mctp^{OG9}$ (red). (C) Average EPSC rise time (20–80%) for *wild type* (gray) and *mctp* (red). D) Average EPSC amplitudes . (E) Average cumulative EPSC as in (D). (F) Representative data for the cumulated peak EPSC amplitudes following a train of action potential (60 Hz, 30 stimuli) shown for *wild type* and $mctp^{OG9}$. Data are fit with a linear function at x = 20 to x = 29 and back-extrapolated to time 0, intersecting the y-axis. Student's t-test ** for p<0.01, n.s. for p>0.05.

dynamics were observed in the two additional *mctp* mutant alleles (data not shown). We then compare short-term release dynamics for wild type and *mctp* under $[Ca^{2+}]_e$ where baseline release is equivalent. For example, comparing wild type at 0.5 mM $[Ca^{2+}]_e$ with *mctp* at 0.7 mM $[Ca^{2+}]_e$ it is apparent that baseline EPSCs are equivalent (p>0.05) and so are short-term release dynamics (p>0.05) (*Figure 5A,B*). These data suggest that the change in short-term release dynamics could be due to a decrease in presynaptic release probability in the *mctp* mutant compared to wild type. Regardless of the underlying cause, it is clear that *mctp* is essential for normal baseline neurotransmitter release and short-term synaptic plasticity.

## MCTP acts downstream of calcium influx to promote the fusion of an EGTA-sensitive synaptic vesicle pool

We observe decreased presynaptic release and enhanced facilitation in the *mctp* mutant. These effects could reasonably be caused by disruption of the readily releasable vesicle pool (RRP) or decreased action-potential (AP) induced calcium influx. We tested both. First, we demonstrate that the RRP is unchanged in *mctp*[OG9] (*Figure 5C–F*), estimated following repetitive nerve stimulation at elevated extracellular calcium according to standard methods (*Schneggenburger et al., 1999*; *Müller et al., 2012*). The estimated RRP for wild type is 953 (±56) compared to 1075 (±67) for *mctp* (n.s.; p>0.05, t-test, two tailed). We also demonstrate that there is no change in EPSC rise time (*Figure 5C*) nor is there a change in the cumulative EPSC comparing wild type and the *mctp* mutant (*Figure 5E*). However, if one divides the cumulative EPSC amplitude by the initial EPSC amplitude, it is possible to arrive at an estimate of the initial release probability on the first stimulus relative to the RRP that is calculated from the entire stimulus train, referred to as $P_{train}$. Since the RRP is unchanged comparing wild type and *mctp* but the initial EPSP is smaller in *mctp*, it is evident that $P_{train}$ is reduced in *mctp.* Quantitatively, we find a significant 21% decrease in $P_{train}$ for *mctp* compared to wild type (p<0.01, t-test, two-tailed, N = 10),

Often, a change in presynaptic release probability is associated with a change in presynaptic calcium influx (*Zucker and Regehr, 2002*). Therefore, we assessed action-potential initiated presynaptic calcium influx by loading presynaptic terminals with the calcium indicator Oregon Green BAPTA-1 (OGB-1), according to published methods (*Müller and Davis, 2012*). Surprisingly, we find no difference in the action potential induced, spatially averaged calcium signal comparing wild type and the *mctp*[OG9] mutant (*Figure 6A,B*). This lack of effect was true for single action potential stimulation as well as repetitive stimulation using five action potentials delivered at the same frequency used to document a change in presynaptic facilitation (*Figure 6D*). There was no change in the time constant of OGB-1 signal decay, consistent with the conclusion that the loss of MCTP does not significantly impact presynaptic calcium buffering (*Figure 6C*). Given that we are directly measuring the spatially averaged calcium signal that is believed to influence presynaptic release, the observed difference in action-potential evoked neurotransmitter release and short-term release dynamics in *mctp* cannot be accounted for by a change in presynaptic calcium influx, nor can it be accounted for by altered calcium efflux from the presynaptic ER. These data demonstrate that MCTP must act downstream of presynaptic calcium influx in order to influence synaptic vesicle release and short-term release dynamics.

A remaining possibility is that MCTP is necessary for efficient coupling of calcium to vesicle release. To probe this possibility, we measured release in the presence of a slow calcium buffer. EGTA has a relatively slow calcium-binding rate (*Smith et al., 1984*) and has been used to probe the functional coupling of vesicle to sites of presynaptic calcium influx (*Schneggenburger and Neher, 2005*). We tested the effects of EGTA-AM (25 μM or 125 μM) on single AP-evoked EPSCs and short-term release dynamics at 1.5 mM $[Ca^{2+}]_e$ comparing wild-type and *mctp*[OG9] mutant synapses (*Figure 7*). At the wild type synapse, the presence of EGTA-AM causes a significant decrease in EPSC amplitude. In contrast, EPSCs in *mctp*[OG9] mutants are completely insensitive to EGTA-AM (*Figure 7B*). We note that EGTA-AM diminishes wild type EPSC amplitudes to precisely the levels observed in the *mctp* mutant in the absence of EGTA-AM. Since *mctp* is insensitive to EGTA-AM, we conclude that MCTP acts downstream of calcium influx to facilitate the release of an EGTA-sensitive vesicle pool during single action potential stimulation.

During repetitive stimulation, intra-terminal calcium levels rise, which can enhance the recruitment of weakly coupled vesicles (*Figure 7*). In the *mctp*[OG9] mutant, access to weakly coupled vesicles is impaired. Our estimates suggest that the RRP in *mctp* is normal, an effect that can be accounted for

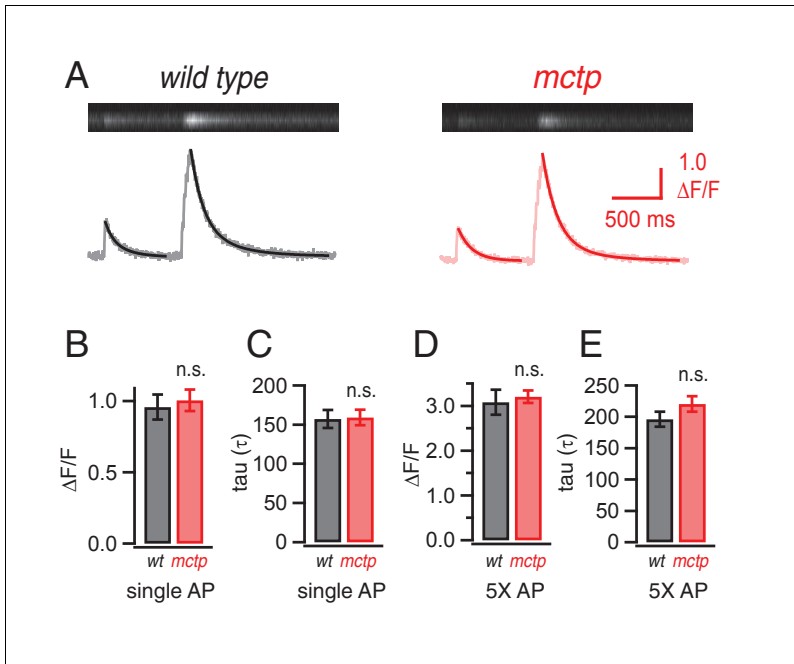

**Figure 6.** Action-potential induced calcium influx is normal in the *mctp* mutant. (**A**) Representative line-scan images at a single synaptic bouton revealing the spatially averaged calcium transients following a single action potential (first stimulus in trace) and a short train of action potentials (5 stimuli at 50 Hz, second stimulus in trace) for wild type (black) and *mctp*$^{OG9}$ (red). Below the raw line-scan data are values (ΔF/F) reporting the change in OGB-1 fluorescence relative to baseline as a function of time. Trials were separated by 750 ms. (**B**) Average peak amplitude of calcium transients (ΔF/F) for single action potential (single AP). (**C**) Average decay (tau) for a single action potential for wild type (n = 10 boutons) and *mctp*$^{OG9}$ (n = 12 boutons). (**D**) Average peak amplitudes of calcium transients (ΔF/F) as in (**B**) for the short stimulus train (5X AP). (**E**) Average decay (tau) as in (**C**) for trains of five action potentials (5X AP).

by pronounced facilitation during the early phase of the stimulus train (*Figure 7*). Indeed, repetitive stimulation reliably invokes facilitation in the *mctp* mutant. One interpretation is that the elevated intra-terminal calcium that occurs during repetitive stimulation overcomes impaired access to the weakly coupled vesicle pool caused by loss of MCTP. EGTA should buffer the slow rise in intra-terminal calcium during a stimulus train and, thereby, the calcium-dependent recruitment of weakly coupled vesicles. Thus, EGTA should convert facilitation to depression in *mctp* mutants. This is precisely what we observe. In wild type, application of EGTA-AM has little effect on short-term release dynamics. However, in *mctp*, short-term facilitation is converted to short-term depression (*Figure 7A,B*). Thus, we provide evidence that MCTP is necessary to promote the fusion of a sub-population of synaptic vesicles that are weakly coupled to sites of action potential induced presynaptic calcium entry.

Finally, we asked whether the role of MCTP during PHP relies upon expansion of the MCTP-dependent, EGTA-sensitive vesicle pool. This appears not to be the case. In the presence of 25 μM EGTA-AM, the induction of PHP occurs normally (*Figure 7C*). Next, we increased the concentration of EGTA five-fold (125 μm), keeping the incubation time for EGTA-AM constant. In this condition, baseline release is significantly impaired, reducing quantal content by ~50%. None-the-less, we observe statistically significant PHP, although it is suppressed relative to controls in the absence of EGTA-AM (*Figure 7D*). Thus, PHP does not require the MCTP-dependent, EGTA-sensitive vesicle pool. MCTP appears to have an additional activity that is necessary for PHP.

## MCTP ensures robust expression of PHP

We next considered the effects of external calcium on the expression of PHP. In wild type animals, PHP is fully expressed across a >15-fold change in extracellular calcium, from 0.3 mM to 15 mM

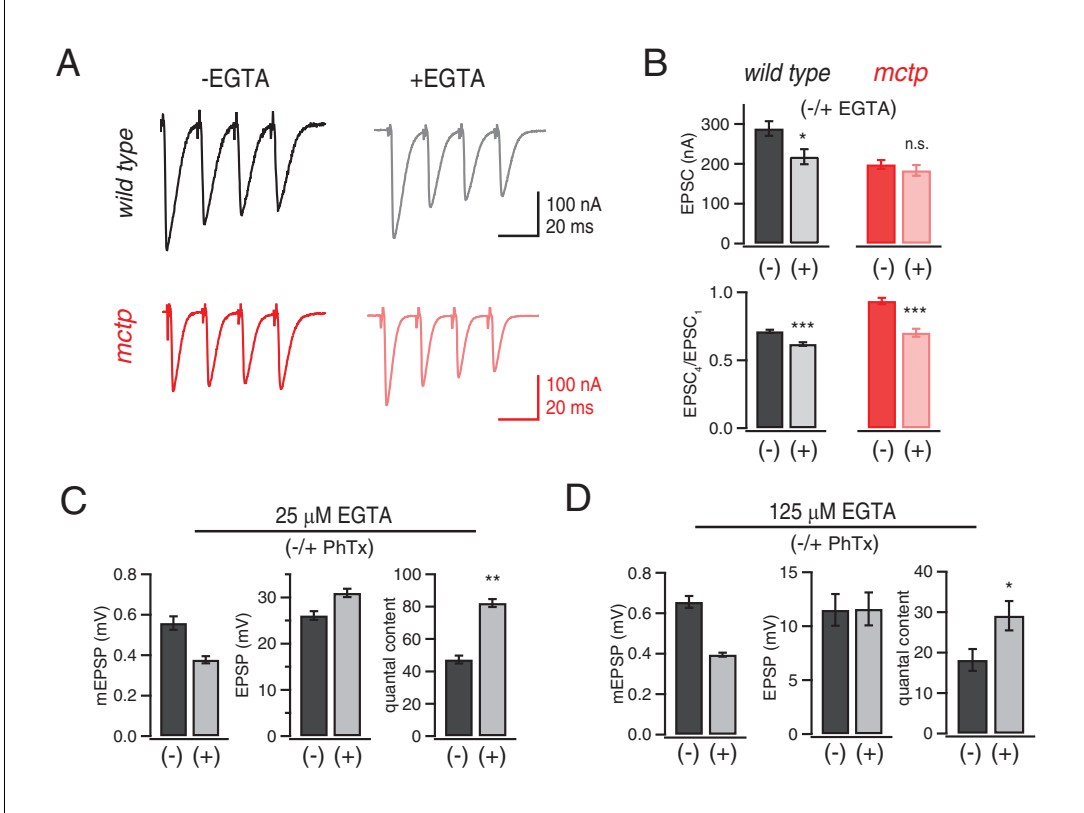

**Figure 7.** Evidence that MCTP is necessary for the release of an EGTA-sensitive pool of synaptic vesicles. (A) Representative EPSC traces shown for wild type and $mctp^{OG9}$ (black and red , respectively) in the absence and presence of EGTA-AM as indicated. (B) Average EPSC amplitudes and short-term modulation of EPSC amplitude (fourth EPSC / first EPSC) for wild type and $mctp^{OG9}$ in the presence (+) or absence (−) of EGTA as indicated (sample size for *wild type*: (-EGTA) n = 12, (+EGTA) n = 11; for *mctp*: (-EGTA) n = 10, (+EGTA) n = 10). (C) Average mEPSP amplitude, EPSP amplitude and quantal content for wild type in the presence (+) or absence (−) of 25 µM EGTA, as indicated. Sample sizes as follows: -PhTx, n = 7; +PhTx, n = 4. (D) Recordings as in (C) performed in the presence of 125 µM EGTA-AM and the presence or absence of PhTx as indicated. Sample sizes as follows: -PhTx, n = 13; +PhTx, n = 15.

(*Müller et al., 2015*). This is particularly remarkable when one considers that PHP is an accurate form of compensation, precisely offsetting the postsynaptic impairment of glutamate receptor function. In other words, PHP can precisely offset a 50% reduction in glutamate receptor function over a >15-fold range of extracellular calcium. There are many non-linear process that are combined in the calcium-coupled release process, particularly over such a large range of extracellular calcium (*Zucker and Regehr, 2002*). Therefore, it is truly remarkable that homeostatic plasticity can remain a constant fraction of total release over this calcium range.

We assayed PHP over a range of extracellular calcium (0.3 mm to 1.5 mM $[Ca^{2+}]_e$), comparing wild type and $mctp^{OG9}$ mutants using two-electrode voltage clamp measurement of synaptic currents. Consistent with prior evidence, in wild type animals, PHP is normally expressed at all extracellular calcium concentrations tested (*Figure 8*). However, *mctp* mutants revealed calcium sensitive expression of PHP. The physiological range for extracellular calcium at the *Drosophila* NMJ is considered to be 0.7–2.2 mM (*Stewart et al., 1994*). At low extracellular calcium (0.3 mm and 0.5 mM $[Ca^{2+}]_e$), PHP is completely blocked in $mctp^{OG9}$ (*Figure 8*, see also *Figures 1* and *2*). When $[Ca^{2+}]_e$ is elevated into the physiological range (0.7 mM and 1.0 mM $[Ca^{2+}]_e$), PHP remains blocked (*Figure 8C*). Only when $[Ca^{2+}]_e$ is increased into the higher end of the physiological range (1.5 mM $[Ca^{2+}]_e$) is PHP expressed in $mctp^{OG9}$. We propose that MCTP normally functions to ensure that PHP is robustly and accurately expressed under variable $[Ca^{2+}]_e$ conditions. To do so, MCTP must have a profound ability to counter changes in extracellular calcium and ensure robust doubling of presynaptic release in the presence of PhTx.

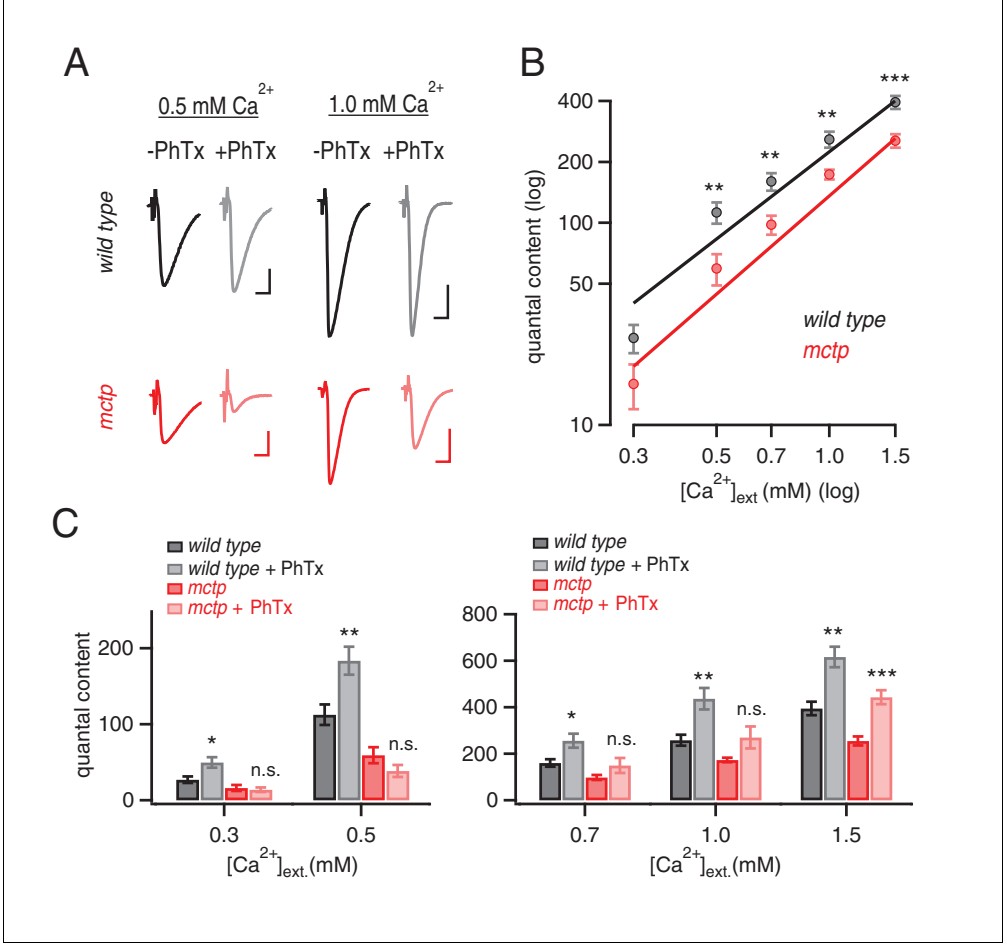

**Figure 8.** MCTP ensures that PHP is robust to variations in extracellular calcium. (**A**) Representative EPSC traces for wild type (black) and $mctp^{OG9}$ (red) at the indicated extracellular calcium concentrations, in the absence (−) and presence (+) of PhTx. Scale bar: 20 nA, 5 ms. (**B**) Average quantal content quantified at indicated calcium concentrations for wild type (black) and $mctp^{OG9}$ (red). (**C**) Average quantal content for wild type (black/gray) and $mctp^{OG9}$ (red/light red) in the absence and presence of PhTx at the indicated extracellular calcium concentrations. Sample sizes are shown for each calcium concentrations (mM) 0.3, 0.5, 0.7, 1.0 and 1.5, respectively: wild type (-Phtx) n = 10, 13, 6, 16, 12; wild type (+Phtx) n = 9, 12, 9, 7, 8. mctp (-Phtx) n = 8, 10, 14, 12, 9; mctp (+Phtx) n = 8, 10, 9, 6, 10. Statistical significance as indicated: Student's t-test * for p<0.05; ** for p<0.01, n.s. for p>0.05.

## Normal active zone ultrastructure in *mctp* mutants

It remains possible that the effects of MCTP on baseline release and PHP could be mediated by changes in the number or distribution of synaptic vesicles at individual active zones. To address this possibility, we visualized active zones using transmission electron microscopy according to published methods (*Harris et al., 2015*). We find no change in active zone ultrastructure or vesicle distribution in *mctp* (*Figure 9*). There is no change in vesicle diameter, indicative of normal vesicle recycling at this synapse (*Poskanzer et al., 2006*). There is no change in the average number of vesicles per active zone, nor is there a change in the average vesicle distance from the center of the presynaptic T-bar, the site where presynaptic CaV2.1 calcium channels reside (*Figure 9D–F*). These data are consistent with our electrophysiological data revealing a normal RRP in the *mctp* mutant (*Figure 5*). Finally, there is no change in the absolute length of the active zone, defined as the electron dense pre- and postsynaptic membranes that flank either side of a T-bar (*Harris et al., 2015*) (*Figure 9C*). Thus, under these conditions, altered release is not correlated with a change in presynaptic ultrastructure in *mctp*. Again, MCTP must have an additional action, downstream of calcium influx that is necessary for normal PHP and baseline release.

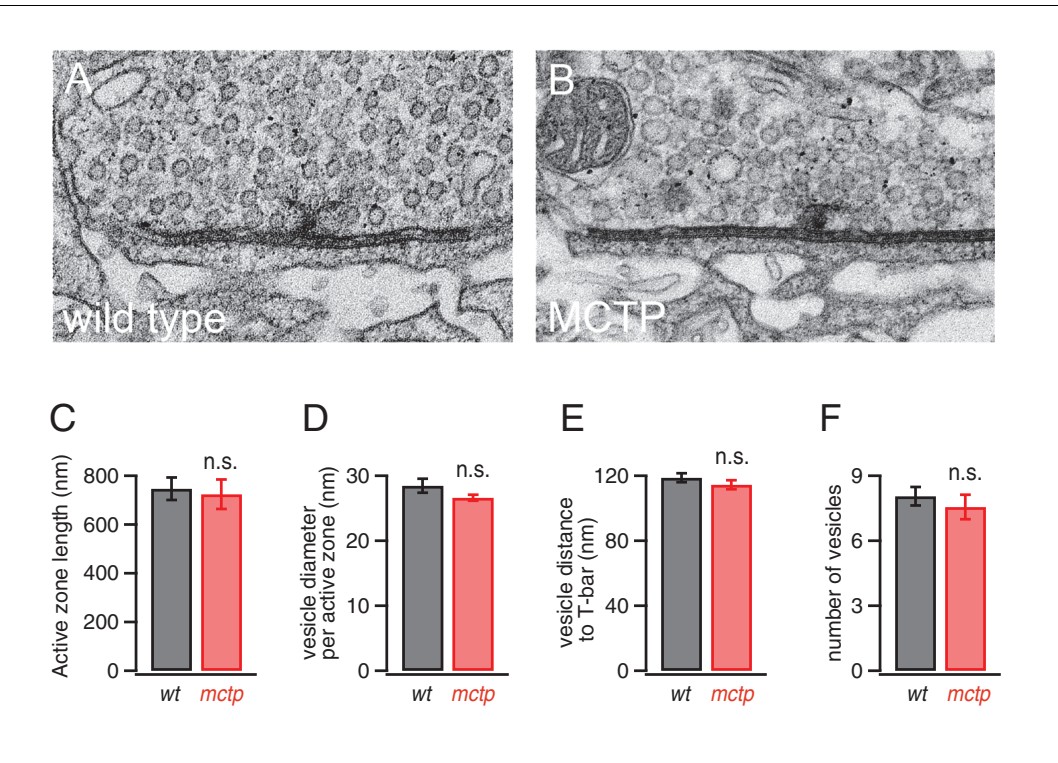

**Figure 9.** Normal active zone ultrastructure in the *mctp* mutant. (A–B) Representative electron microscopy images of an active zone from *wild type* (A) and *mctp*$^{OG9}$ (B). (C–F) Average active zone length (C), vesicle diameter per active zone (D), average vesicle distance to T-bar (E) and the average number of vesicles within 150 nm of the presynaptic T-bar (F). Sample sizes as follows: number of active zones in *wild type*, n = 19; *mctp*, n = 16; sections from at least two NMJ in two separate animals. Student's t-test * for p<0.05; ** for p<0.01, n.s. for p>0.05.

## Evidence that MCTP is an ER calcium sensor that drives expression of PHP

It was previously demonstrated that all three C2 domains of MCTP bind calcium (*Shin et al., 2005*). To determine whether the calcium-coordinating residues in C2A, C2B and C2C domains are necessary for PHP, we mutated all of the predicted calcium coordinating residues from aspartic acid to asparagine. There are five critical residues predicted to be necessary for calcium coordination in C2A, two in C2B and five in C2C. Four mutant transgenes were generated. The first transgene included mutations in all twelve residues, termed *mctp*$^{C2D12N}$, which is predicted to render the MCTP protein unable to coordinate calcium in any of the three C2 domains. Then, we generated transgenes in which single C2 domains were mutated, termed *mctp*$^{C2A}$*, *mctp*$^{C2B}$* and *mctp*$^{C2C}$*. These transgenes were expressed under *UAS*-control in neurons (*elav-Gal4*) in the *mctp*$^{OG9}$ mutant background. Each transgene was epitope tagged, allowing us to follow the distribution of the over-expressed protein to ensure proper trafficking and localization of the mutant transgenes compared to the wild type epitope tagged transgene. Each transgene was confirmed by direct sequencing.

We find that neuronal expression of *UAS-mctp*$^{C2D12N}$ fails to rescue PHP in the *mctp*$^{1}$ mutant and does not impair release beyond that observed in the *mctp*$^{1}$ mutant alone (data not shown). Given this, we tested the transgenes harboring mutations in individual C2 domains. First, we demonstrate that each *mctp* transgene (*mctp*$^{C2A}$*, *mctp*$^{C2B}$* and *mctp*$^{C2C}$*) is well expressed and is distributed throughout the presynaptic terminal (*Figure 10A*). Compared to expression of wild type *mctp-flag* (*Figure 10A* top left), expression of both *mctp*$^{C2B}$*-*flag* and *mctp*$^{C2C}$*-*flag* show normal abundance and distribution within the presynaptic terminal. Expression of *mctp*$^{C2A}$*-*flag* is qualitatively less abundant, but it does traffic to the presynaptic terminal where it appears to be normally distributed. All transgenes are inserted at a common genomic site, suggesting that these differences in protein

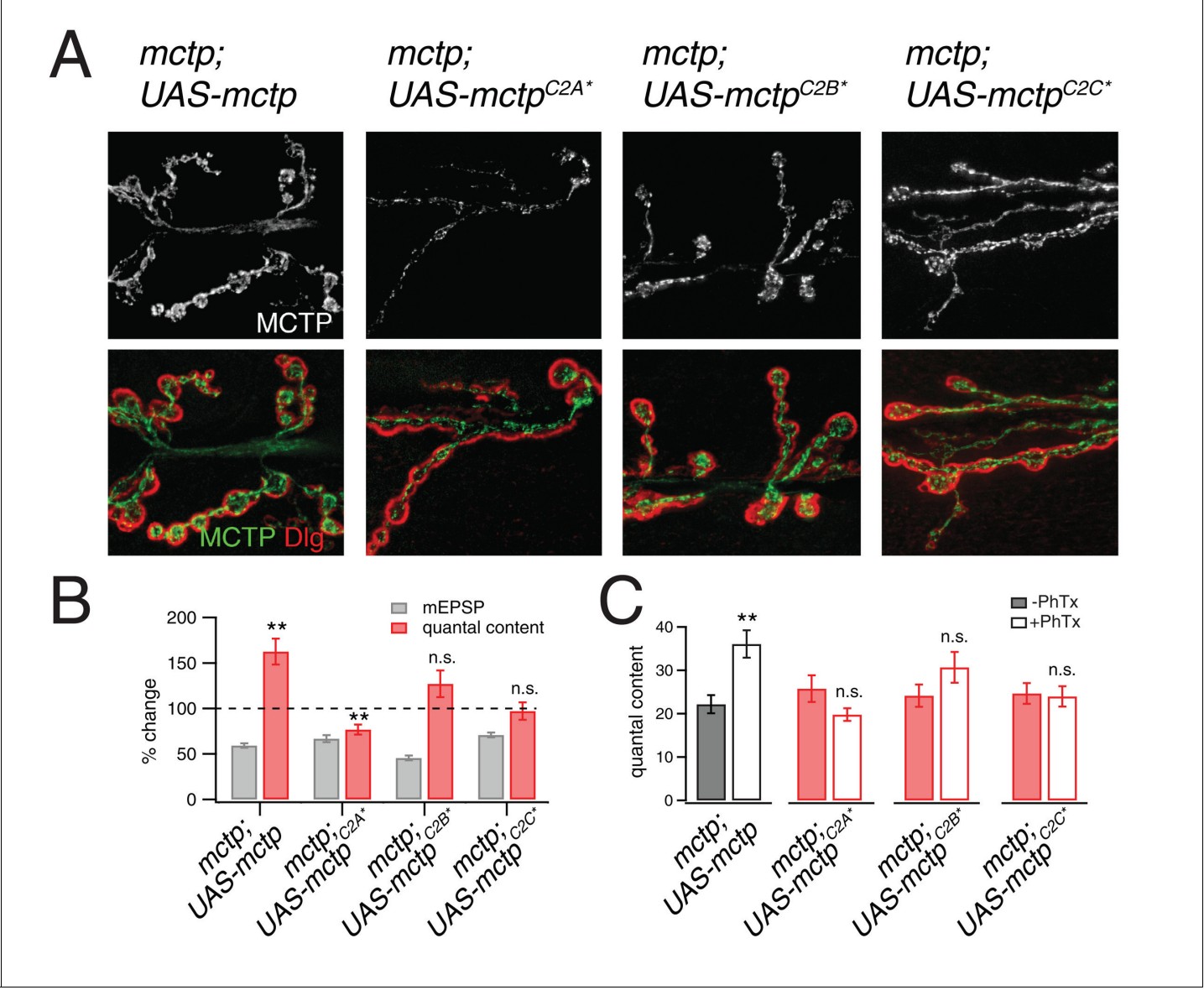

**Figure 10.** Evidence that MCTP is an ER localized calcium sensor. (A) Representative images of the NMJ showing the expression of epitope tagged *mctp* transgenes (*UAS-mctp-Flag*). At top (black and white) are images of MCTP-Flag at the NMJ. Below, staining is shown for MCTP (green) and anti-Dlg (red). Sequentially, moving to the right are images for expression of mutant *UAS-mctp-flag* transgenes in the *mctp*[OG9] mutant background. Each transgene harbors mutations in an individual C2 domain, either in the C2A domain (C2A*), the C2B domain (C2B*) or the C2C domain (C2C*, at far right). In all cases, the *elav-Gal4* driver is used to express transgenes in neurons in the *mctp*[OG9] mutant. (B) Average percent changes in mEPSP amplitude (filled bars) and quantal content (open bars) for the indicated genotypes (*mctp, mctp*[C2A*], *mctp*[C2B*], *mctp*[C2C*]) in the presence of PhTx compared to baseline. (C) Quantification of mEPSP amplitudes and quantal content for mctp (-PhTx, n = 10; +PhTx, n = 10), *mctp*[C2A*] (-PhTx, n = 14; +PhTx, n = 8), *mctp*[C2B*](-PhTx, n = 6; +PhTx, n = 9), *mctp*[C2C*] (-PhTx, n = 14; +PhTx, n = 10). Student's t-test ** for p<0.01, * for p<0.05, n.s. for p>0.05.

distribution are not caused by differences in transgene expression. None-the-less, since all three transgenes are present at the NMJ, it argues that these transgenic proteins do not dramatically alter the ER membrane system and the extension of this membrane system to the presynaptic terminal. With this information in hand, we tested PHP and find that all three transgenes fail to rescue PHP in the *mctp*[OG9] mutant background (*Figure 10B,C*). There is no increase in quantal content in *mctp* mutants expressing either *UAS-mctp*[C2A*] or *UAS-mctp*[C2C*]. There is a trend toward increased quantal content in animals expressing *UAS-mctp*[C2B*], but this fails to reach significance. We conclude

that MCTP functions as an ER-resident calcium sensor that is necessary for PHP. Since PHP requires calcium-coordinating residues in all three MCTP C2 domains, we propose that MCTP functions as a presynaptic calcium sensor and a source of calcium-feedback signaling to ensure robust expression of PHP.

## Discussion

Here we provide evidence that MCTP is an ER-resident calcium sensor that promotes the release of a population of weakly coupled synaptic vesicles and drives robust expression of PHP. We demonstrate that loss of MCTP does not alter the spatially averaged presynaptic calcium signal in response to single or repetitive nerve stimulation. Thus, MCTP functions downstream of cytosolic calcium originating from either plasma membrane calcium channels or ER calcium stores. Prior evidence suggests that MCTP is sensitive to calcium in the 1 µM range (*Shin et al., 2005*). As such, MCTP could 'sense' action-potential induced calcium influx or residual calcium following an action potential. Prior evidence also indicates that MCTP does not bind membrane lipids (*Shin et al., 2005*). This does not rule out the possibility that MCTP could confer calcium sensitivity to the protein complexes that mediate an interaction of ER membranes with plasma membranes at or near presynaptic release sites. A recent paper has underscored the role of axonal ER in the regulation of presynaptic release (*de Juan-Sanz et al., 2017*). In this work, depletion of ER calcium strongly suppresses presynaptic release in a STIM1-dependent manner (*de Juan-Sanz et al., 2017*). However, STIM-1 knockdown was shown to have no effect on baseline transmission under normal physiological conditions (*de Juan-Sanz et al., 2017*). By contrast, our work specifies a role for ER-localized MCTP as a critical regulator of baseline release, short-term plasticity and PHP, under physiological conditions.

### Separable actions of MCTP support baseline release and PHP

Loss of MCTP appears to have two separable effects on presynaptic release. On the one hand, loss of MCTP impairs presynaptic vesicle release across a 3-fold range of extracellular calcium (0.5–1.5 mM $[Ca^{2+}]_e$). This represents a shift in the calcium sensitivity of release without a change in the calcium-dependent cooperativity of release. We further demonstrate that loss of MCTP causes increased short-term facilitation. This effect is generally associated with decreased presynaptic release probability. Indeed, there is no change in stimulus-dependent RRP size or synaptic ultrastructure. However, there is also no change in the spatially averaged calcium signal in response to single or repetitive action potential stimulation. Rather, it appears that MCTP controls access to an EGTA-sensitive component of presynaptic release, also termed 'weakly calcium-coupled vesicles'. It is worth noting that the magnitude of the decrease in baseline release in *mctp* under physiological calcium conditions is quantitatively similar to that observed in mutations deleting Rab3-Interacting Molecule (RIM; *Müller et al., 2012*), a presynaptic component of the active zone cytomatrix (*Südhof, 2012*). As such, the effects of *mctp* mutants are dramatic and consistent with emerging evidence that presynaptic ER signaling is a potent regulator of synaptic transmission (*de Juan-Sanz et al., 2017*).

MCTP is also required for the expression of PHP. Unlike the effects on baseline release, which occur over a range of 0.5 to 1.5 mM $[Ca^{2+}]_e$, the block of PHP in the *mctp* mutant occurs when extracellular calcium concentration is diminished below 1.5 mM $[Ca^{2+}]_e$. The physiological range for extracellular calcium at the *Drosophila* NMJ is considered to be 0.7–2.2 mM (*Stewart et al., 1994*). It is remarkable, therefore, that PHP can be both accurate and robust when recordings are made at 0.3 mM $[Ca^{2+}]_e$ at a wild type synapse. One possibility is that MCTP functions as part of a calcium-dependent feedback signaling system that stabilizes the expression of PHP. Since PHP can be induced in the absence of action potential induced presynaptic calcium influx (*Frank et al., 2006*), MCTP would need to be sensitive to changes in extracellular calcium, presumably detecting corresponding changes in cytoplasmic calcium since MCTP resides on the internal ER membranes. In the mammalian central nervous system, there is considerable evidence that extracellular calcium concentrations can be substantially diminished, well below the physiological range (hypocalcaemia) during periods of normal neural activity or as a consequence of disease (*Lu et al., 2010*; *Borst and Sakmann, 1999*; *Silver and Erecińska, 1990*). As such, MCTP-dependent stabilization of PHP could be an essential mechanism to prevent the failure of synaptic transmission at neuromuscular and, possibly, central synapses under convergent conditions of hypocalcaemia and homeostatic stress.

## Potential models for MCTP-mediated stabilization of PHP

One possibility is that MCTP-dependent signaling occurs at sites of ER-plasma membrane contact. This is consistent with the demonstration that the ER-plasma membrane interface is controlled in a calcium-dependent manner in other cell types. For example, the eSyts (*Idevall-Hagren et al., 2015*; *Saheki et al., 2016*; *Giordano et al., 2013*) and orthologous Tricalbins in yeast (*Manford et al., 2012*; *Toulmay and Prinz, 2012*) are calcium sensitive molecular tethers that control the extent of ER-plasma membrane contact. One action of the eSyts is to control plasma membrane lipid homeostasis through the action of their SMP domain (*Saheki et al., 2016*; *Schauder et al., 2014*). This represents a homeostatic system that controls plasma membrane phospholipid composition (*Saheki et al., 2016*). MCTP lacks an SMP domain, but this does not rule out the possibility that MCTP functions as part of a related protein complex that mediates such an activity. As another example, it has been established that oligomerization of the ER-resident protein STIM1 occurs in response to lowered levels of calcium within the ER, triggering its interaction with and activation of the plasma membrane calcium channel Orai1. This represents a calcium-dependent signaling system that sustains calcium levels in the ER lumen (*Feske et al., 2006*; *Zhang et al., 2005*; *Liou et al., 2005*) and has been recently linked to the regulation of presynaptic release under conditions of ER calcium depletion (*de Juan-Sanz et al., 2017*). It is worth noting that ER-plasma membrane contact sites exist prior to calcium-dependent STIM1 recruitment, an observation that has highlighted the importance of ER-plasma membrane contact sites as platforms for other signaling events (*Giordano et al., 2013*; *Anderie et al., 2007*; *Stefan et al., 2011*, *2013*). It is also worth noting that proteins resident at ER-plasma membrane contacts are not always selectively localized to these sites (*Giordano et al., 2013*), an observation that is consistent with the observed distribution of MCTP throughout the ER in neurons.

It remains to be determined whether loss of MCTP disrupts ER-plasma membrane contacts in the presynaptic terminal and whether these hypothetical sites are controlled in a calcium dependent manner via MCTP activity. Indeed, the observation that C2-domain specific mutations in MCTP block PHP without dramatically altering the distribution of MCTP-labeled ER membranes within the presynaptic terminal argues against the simple possibility that MCTP controls PHP by organizing ER morphology. It seems more likely that MCTP could confer calcium-dependence to signaling between the ER and other internal membrane systems. Without question, there are many experiments that could be performed to pursue these ideas. None-the-less, we provide strong evidence that MCTP is a calcium sensor that acts downstream of calcium influx to stabilize baseline release and confer robustness to the expression of PHP.

## Relevance to disease

Disruption of the extended neuronal endoplasmic reticulum has been repeatedly associated with neurological disease. For example, the spastic paraplegias are genetic disorders that include degeneration of upper motoneurons (*Blackstone et al., 2011*). Among the more than 70 gene mutations responsible for this disease are mutations in the gene encoding ER-resident proteins Atlastin1 and Reticulon2. A recent study demonstrated that RNAi-mediated knockdown of either gene in *Drosophila* motoneurons causes ER fragmentation, decreased neurotransmitter release and altered synaptic morphology (*Summerville et al., 2016*). Other recent studies have examined mutations in *Drosophila spastin*, *spartin* and *spichthyin*, all of which disrupt the integrity of the ER membrane system (*Nahm et al., 2013*; *Ozdowski et al., 2011*; *Sherwood et al., 2004*; *Wang et al., 2007*). Each of these mutations also disrupts synaptic morphology, alters the microtubule cytoskeleton and cause defects in synaptic transmission. There are notable and important differences compared to our observations of *mctp* mutants. First, homeostatic plasticity was not tested in these prior studies, so a direct comparison cannot be made. Second, we did not observe changes in ER organization in the *mctp* mutant and the microtubule cytoskeleton is unaltered in *mctp*. Third, there is no substantive change in NMJ growth, organization or changes in synaptic ultrastructure caused by loss of *mctp*. Finally, we extend our work to the analysis of calcium-coordinating residues in the individual C2 domains of MCTP and demonstrate that the distribution of C2-domain mutations are largely normal, implying that PHP is blocked while the ER membrane system remains intact. In this respect, it is very intriguing that mutations in human *mctp* were recently associated with psychiatric disease in humans (*Djurovic et al., 2009*; *Scott et al., 2009*). Bipolar disorder can be considered a disease of neuronal

malfunction that could reasonably be related to the role of MCTP in the homeostatic stabilization of synaptic transmission.

## Materials and methods

### Fly stocks

*Drosophila* stocks were raised at 25°C on standard food. The $w^{1118}$ strain is used as the wild-type control unless otherwise noted because this is the genetic background of the transgenic lines used. Both $mctp^1$ and $mctp^2$ were obtained from the Bloomington Drosophila Stock Center (Bloomington, IN). $mctp^1$ was the initial mutant isolated from the synaptic homeostasis screen (*Dickman and Davis, 2009*), and is a piggybac transposon insertion ($Pbac^{e00415}$) residing in the 3' UTR. $mctp^2$ is a minos transposon insertion (*MI102901*) in a coding exon of the *mctp* gene. All other *Drosophila* stocks were obtained from the Bloomington Drosophila Stock Center unless otherwise noted. For pan-neuronal expression of *mctp* transgenes and presynaptic rescue, we used the $elav^{c155}$-*Gal4* driver. The myosin heavy chain driver (*MHC-Gal4*) was used for postsynaptic rescue. Standard second and third chromosome balancers and genetic strategies were used for all crosses and for maintaining mutant lines.

### Immunochemistry

Third-instar larvae were dissected, fixed in Bouin's fixative or 4% PFA in PBS, and immunostained as previously described (*Eaton et al., 2002*; *Harris et al., 2015*). Dissected third instar larvae were fixed with PFA (4%) and incubated overnight at 4 C with primary antibodies (mouse anti Flag 1:50; rabbit anti-RFP 1:100; rabbit anti-Dlg, 1:1000; anti-Syt1 1:1000, anti-Brp 1:100, Life Technologies). Alexa-conjugated secondary antibodies and goat anti-HRP were used (Jackson Laboratories 1:500). Images were acquired with either a Zeiss LSM700 confocal microscope equipped with Zen software using a 63X 1.6 NA oil immersion objective or an upright epifluorescence deconvolution confocal microscope (Axiovert 200, Zeiss) equipped with a 100X objective (N.A. 1.4), cooled CCD camera (CoolSnap HQ, Roper Scientific). Slidebook 5.0 (3I, Intelligent Imaging) was used for capturing, deconvolving and analyzing images. Structured illumination microscopy (Nikon LSM 710 equipped with 63X objective and Andor Ixon EMCCD camera) was used to perform Brp-GFP and MCTP-Flag colocalization experiments. Bouton numbers and Brp numbers and densities were quantified as described previously (*Harris et al., 2015*).

### Electrophysiology

Current clamp and two-electrode voltage clamp recordings were done from muscle six, at the second and third segment of the third-instar *Drosophila* with Axon 2B or an Axoclamp 900 amplifiers (Molecular Devices). The extracellular solution (HL3) included (in mM) 70 NaCl, 5 KCl, 10 $MgCl_2$, 10 $NaHCO_3$, 115 sucrose, 4.2 trehalose, 5 HEPES. $Ca^{2+}$ was added to the HL3 at the day of the experiment at different concentrations (0.3 to 1.5 mM). Homeostatic plasticity was induced by incubating the larvae with Philantotoxin-433 (PhTx, 15–20 μM, Sigma) for 10 min as previously described (*Frank et al., 2006*; *Harris et al., 2015*). EGTA-AM was diluted in calcium-free HL3 to its final concentration (25 and 125 μM, Sigma), applied to the unstretched larvae for 10 min of incubation. Quantal content was calculated by dividing average EPSP to mEPSP. For estimating the pool size, cumulative peak EPSP amplitudes at 60 Hz were line-fitted and back-extrapolated to time = 0 (*Schneggenburger et al., 1999*). mEPSPs were analyzed with MiniAnalysis program (Synaptosoft). All other physiology data were analyzed with custom written functions in Igor 6 (Wavemetrics Inc).

### Calcium imaging

Calcium imaging was performed as previously described (*Müller and Davis, 2012*; *Müller et al., 2015*). Final stocks of OGB-1 488 (1 mM, Sigma) and Alexa-568 (5 mM, Sigma) were prepared in HL3 (0 mM $Ca^{2+}$). Third instar *Drosophila* larvae was dissected and incubated on ice for 10 min in HL3 with zero calcium (1 mM OGB-1; 5 mM Alexa 488, Invitrogen). Indicators were removed and larvae were washed for 10 min with HL3, then placed into the recording chamber for imaging. A scanning confocal microscope (Ultima, Prairie Technologies) with a 60× objective (1.0 NA, Olympus) was used for imaging. 488 nm excitation wavelength from a krypton-argon laser used for excitation and

emitted photons were collected through a pinhole at a photocathode photomultiplier tube (Hamamatsu). All line scans were performed at type 1b boutons of muscle 6/7, segments A2-A3. Loading efficiency of the dye was assessed by the intensity of co-loaded Alexa 568. Single stimuli (1 ms) and stimulus trains (5 pulse, 1 ms duration, 50 Hz) were used. Changes in the fluorescence were quantified as previously described (*Müller and Davis, 2012*; *Müller et al., 2015*).

## Electron microscopy

Electron microscopy was performed as previously described (see *Harris et al., 2015*).

## Molecular biology

Two partial *mctp* cDNA sequences were obtained as cDNA clones from the Berkeley Drosophila Research Consortium, IP11216 and RE18318 (www.bdgp.org). Standard PCR methods were used to amplify overlapping cDNA sequences from these clones. These sequences were then used in a duplex PCR reaction to generate the full length *Drosophila mctp* open reading frame. This full length *mctp* cDNA was then cloned into the pENTR vector (Gateway Technology, Invitrogen) by engineering a CACC sequence in the forward primer (caccATGTCACGCATCCAATACGTTG) and using a reverse primer, with or without a stop codon (CTATGATCCTTTCAGTTTCTTCTT, TGATCCTTTCAG TTTCTTCTTGG) to amplify the full length *mctp* open reading frame, which was used in subsequent Gateway reactions. To generate the C2 domain mutations, we used PCR site-directed mutagenesis. Amino acids 233, 239, 286, 288, and 294 were changed from D to N in $mctp^{C2A}$. Amino acids 387 and 393 were changed from D to N in $mctp^{C2B}$. Amino acids 534, 540, 586, and 588 were changed from D to N and amino acid 594 was changed from E to N in $mctp^{C2C}$. All constructs were cloned into the proper destination vector (*pUASt-mctp* alone or *pUASt-mctp-3xFlag*) obtained from the *Drosophila* Gateway Vector Collection (Carnegie Institution, Baltimore, MD), and sequenced. Targeted insertion of the mutant transgenes into the *Drosophila* genome (PhiC31 integrase-mediated injection) was achieved using the services of BestGene, Inc (Chino Hills, CA). For Crispr gRNA flies, three gRNA sequences were selected using the CRISPR optimal target finder website (http://tools. flycrispr.molbio.wisc.edu/targetFinder). These sequences were cloned into the pCDF3-dU6:3 gRNA vector (Addgene plasmid #49410, Simon Bullock). Transgenic animals harboring the gRNA vector were generated using standard PhiC31 integrase-mediated injection methods (BestGene). These flies were used to generate mutations in the *mctp* gene locus in combination with transgenically expressed Cas9 at the germline (*Kondo and Ueda, 2013*). Single lines were generated and sequenced. A single mutation was selected ($mctp^{OG9}$), described in the main text.

## Drosophila S2 cells and ER imaging

*Drosophila* S2 cells (obtained from the Vale laboratory, UCSF with additional source information at http://flybase.org/reports/FBtc9000006.html) were grown in Schneider's *Drosophila* medium (Life Technologies) and transfected by Effectene (Qiagen). After 48 hr, cells were plated onto the coverslip coated with ConA. For live ER imaging, cells were incubated with red ER-Tracker (Thermofisher) for 30 min and washed with HL3. Images were collected with a Yokagawa CSU22 spinning disk confocal microscope equipped with 100X objective. Cells are mycoplasma negative (tested by MycoAlert, Lonza, 2017).

## Acknowledgements

We thank Tingting Wang, Brian Orr, Ryan Jones, Nathan Harris for comments on a previous version of this manuscript and the Bloomington Drosophila Stock Center and the Iowa Developmental Studies Hybridoma Bank for genetic and antibody reagents. This work was supported by NIH grants NINDS RO1 NS39313 and R35 NS097212 to GWD as well as NIH grant (MH 092351) and a New Scholar Award from the Ellison Medical Foundation to DKD.

# Additional information

## Competing interests

GWD: Reviewing editor, *eLife*. The other authors declare that no competing interests exist.

## Funding

| Funder | Grant reference number | Author |
| --- | --- | --- |
| National Institute of Neurological Disorders and Stroke | NS039313 | Graeme W Davis |
| National Institute of Neurological Disorders and Stroke | NS097212 | Graeme W Davis |

The funders had no role in study design, data collection and interpretation, or the decision to submit the work for publication.

## Author contributions

ÖG, Conceptualization, Data curation, Formal analysis, Writing—review and editing; DKD, Conceptualization, Formal analysis, Supervision, Funding acquisition, Writing—review and editing; WM, AT, Data curation, Formal analysis; RDF, Data curation, Formal analysis, Methodology, Writing—review and editing; GWD, Conceptualization, Data curation, Formal analysis, Funding acquisition, Methodology, Writing—original draft, Project administration, Writing—review and editing

## Author ORCIDs

Dion K Dickman, http://orcid.org/0000-0003-1884-284X
Graeme W Davis, http://orcid.org/0000-0003-1355-8401

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
