## [Decision Letter]

Thank you for submitting your article "MCTP is an ER-Resident Calcium Sensor that Stabilizes Synaptic Transmission and Homeostatic Plasticity" for consideration by *eLife*. Your article has been reviewed by three peer reviewers, and the evaluation has been overseen by a Reviewing Editor and Richard Aldrich as the Senior Editor. The following individual involved in review of your submission has agreed to reveal his identity: Bing Zhang (Reviewer #2).

Based on the reviews as well as subsequent discussions among the reviewers, we would like to invite you to submit a revised manuscript. From the reviews below, you will see that a common concern is that it is inadequate to determine the subcellular localization of MCTP based on overexpressed proteins. This should be determined by antibody against endogenous protein, or MiMIC or CRISPR/Cas9-produced tag of endogenous protein. We also expect that you will respond to other critiques of the reviewers as thoroughly as possible with new data (e.g., potential artifact of ER localization caused by fixation; testing new Ca^2+^ buffer). Where it is not possible, please tone down the assertions based on available data.

Reviewer #1:

The last 15 years have seen a marked increase in our knowledge of presynaptic homeostatic plasticity (PHP), largely driven by studies from the Davis lab at the fly NMJ. However, the molecular underpinnings of these events are incompletely known. In this manuscript, Genc and colleagues identify MCTP, a C2-domain containing transmembrane protein, as a new player involved in PHP. It is necessary for both acute and chronic invocations of PHP as well as coordination of PHP over a wide range of calcium concentrations. MCTP localizes broadly throughout nerves, and overlaps somewhat with KDEL, suggesting that it localizes (though not exclusively) to the ER. Beyond its involvement in PHP, MCTP also plays a role in baseline neurotransmission, allowing access to a pool of weakly coupled, EGTA-sensitive vesicles, which may also factor into PHP, though the link is somewhat murky. Perturbation of MCTP does not affect morphology or vesicle parameters at the NMJ. Finally, the authors demonstrate that all three C2 domains of the protein are necessary for PHP, suggesting that the calcium-binding activity of MCTP is what drives its role in PHP.

The experiments are well done, the physiology well controlled, and the data (largely, see below) very strong. There is some question, however, as to how the roles of MCTP regarding PHP and the weakly coupled vesicle population are connected and how separable are the functions of MCTP in baseline transmission and PHP. As such, the mechanistic insight offered by this new player into how PHP functions is limited. The feedback system, as suggested by the authors, is a tantalizing prospect, though more insight into that mechanism would be necessary to elevate this paper to *eLife*. This is an excellent analysis of a new PHP player that would be better suited for a specialty journal.

Specific Points:

I am, however, concerned about the conclusions regarding MCTP overexpression. Due to the widespread localization, it is likely to partially colocalize with many markers. So co-staining with Syt, Brp, and HRP is only mildly informative. The KDEL staining is convincing in the segmental nerves (though again, overexpressed), but doesn't match informatively at the NMJ. Thus, I wonder if overexpression causes mislocalization. I would recommend qPCR to compare overexpression levels with those of endogenous. If they are vastly different, identify expression conditions that most closely resemble WT. That would allow for more confidence regarding the localization. Further, commercial KDEL antibodies exist as well as monoclonals to *Drosophila* Golgi and ER proteins (Riedel et al., 2016, Biology Open) that might be helpful to avoid concomitant KDEL overexpression.

In the mctp mutants, the AP waveform is unchanged, but is calcium influx (the currents themselves) is affected? It seems important to know, given the assertions made later on regarding slow calcium buffering with the EGTA experiments.

I'm particularly intrigued by the change from short-term depression to facilitation in mctp in Figure 5. The same change doesn't appear in Figure 7. Rather, it looks like the level is maintained: is the change only present at certain frequencies? Calcium concentrations? The differences could be made clearer. Or perhaps a more representative trace used?

Regarding the interpretation of MCTP function, the presence of PHP at high extracellular calcium levels in mctp is interesting. In this case, the calcium driving force is greater, likely overloading normal buffering, or at least slowing it. This would alter the average extra / intra calcium ratio. Is it possible that MCTP is, instead of detecting extracellular calcium, detecting influx, clearance, or extrusion rates from other buffering mechanisms, like PMCA or SERCA? Calcium imaging of transients at high calcium levels may shed some light on the mechanism by which MCTP might work as a sensor. This could further enlighten a mechanistic role.

Reviewer #2:

This is an interesting and original study on identification of a presynaptic molecule that plays a key role in homeostatic synaptic plasticity. The author identified an ER- multi-domain Ca sensor from a large scale genetic screen and presented strong electrophysiological evidence that MCTP acts downstream of Ca influx to facilitate the expression of presynaptic compensatory increase in quantal release upon a reduction in postsynaptic glutamate receptors. This work extends previous findings from the Davis lab and appears to add a new presynaptic molecule to the network of proteins involved in homeostatic regulation. What makes this report more interesting is that MCTP is a synaptotagmin-like Ca sensor and localized in presynaptic termini, close to the active zones, to regulate presynaptic release. The data presented in this manuscript are solid and the interpretations are reasonable.

1) One of my suggestions is to move Figure 10 to Figure 2 so that the authors can show the structure-function in the context with presynaptic rescue. This is minor.

2) The facilitation experiment comparing EPSP4 /EPSP1 ratio in Figure 5 is consistent with the observation that the basal release at the mctp mutant NMJs is reduced. A better comparison would be to adjust the extracellular [Ca] such that the initial EPSC is similar for both the wild type and mctp mutant.

3) The EGTA experiment is intriguing, and I wonder how does this compare with a fast Ca chelator such as BAPTA-AM?

Reviewer #3:

In this study, Genc and colleagues report the isolation of a putative smooth ER protein, mctp, which when mutated blocks presynaptic homeostatic plasticity. They postulate that mctp is associated with the smooth ER, which has been previously shown to span the entire neuron, from cell body to synaptic terminals to dendrites. They show that the protein is in close proximity to the plasma membrane and sometimes apposed to active zones. In addition, they provide evidence, including mutations in the C2 domains and alterations in calcium levels, that mctp is likely to function downstream of calcium influx into the cell. They propose that mctp is a calcium sensor that serves to stabilize base transmission and homeostatic plasticity as a function of postsynaptic excitability. These observations are significant as we still know little about the mechanisms of homeostatic plasticity.

My main concerns were two-fold, and while they do not detract from most of the main findings in the paper, they do introduce an element of over-interpretation and they should be corrected.

1) As has been previously reported, chemical fixation destroys the integrity of the ER in this system (Summerville et al. 2016). Therefore all statements regarding the localization of the mctp signal near the plasma membrane of the terminals, or the localization at ER regions apposed to active zones is in question. Reagents to look at the ER in unfixed preparations are available.

2) Overexpression of mctp is used to determine the localization of the protein, which is not acceptable, especially given the availability of tools to look at this without overexpression or antibodies (e.g., introduction of a tag into the genomic locus via CRISPR). The arguments presented to argue for overexpression of mctp reflecting the endogenous localization of the protein are not valid as presented (e.g., lack of aggregation of the label might be meaningless as to the normal localization of the protein; that the tagged protein rescues phenotypes can also be meaningless as to protein localization. For example, sufficient amounts of mctp-flag might be present at appropriate locations to rescue abnormal phenotypes, even if mctp-flag overflows to additional sites; the argument that mctp has similar transmembrane domains as another ER protein is also a very weak argument).

3) Another criticism is about the inappropriate interpretation of the statistical analysis. At least in 3 occasions authors point to "a trend (difference in the mean)" even when the statistical analysis says that a difference between the samples cannot be determined at 95% confidence. Further, the authors go on to discuss the significance of the "trend", which is overstretching the data.

[Editors' note: further revisions were requested prior to acceptance, as described below.]

Thank you for resubmitting your work entitled "MCTP is an ER-Resident Calcium Sensor that Stabilizes Synaptic Transmission and Homeostatic Plasticity" for further consideration at *eLife*. Your revised article has been favorably evaluated by Richard Aldrich (Senior editor), a Reviewing editor and two reviewers.

The manuscript has been improved but there are some remaining issues that need to be addressed before acceptance, as outlined below:

As you see, Reviewer #1 raised the caveat of accessing the protein localization purely based on overexpressed protein, which in fact was an issue we highlighted in editorial comments to the original submission. Although your rebuttal cited examples that overexpressed proteins reflect the localization of endogenous proteins, this is not guaranteed for a novel protein. We recognize that the revised manuscript contains a lot of new data, and after discussions among reviewers, we agree that the manuscript is acceptable for publication if you clearly point out the limitations of using overexpressed protein to infer the localization of an endogenous protein.

Reviewer #1:

In this revision, Genc and colleagues have added data regarding the localization of overexpressed MCTP. They conduct co-staining with an overexpressed ER marker (HDEL-GFP) in vivo under live imaging conditions to preserve ER morphology. They also utilize cell culture assays to perform structure-function localization of MCTP. These definitely enhance the paper, and provide further information to support their thesis. Moreover, it provides very useful data regarding the portions of the protein necessary for localization and nicely compliments the analysis regarding the C2 domains and function. I thank them for their additions.

However, they do not address the fundamental criticism that all reviewers raised, in that these are overexpressed proteins. The live imaging improves morphology and resolution while further lending credence to their assertions. And though overexpressed protein CAN localize properly, it is not a definite that it will. The colocalization with HDEL-GFP is very encouraging (though itself overexpressed), the conclusions are limited by not directly assessing the endogenous proteins.

Reviewer #2:

I was the most positive one on the last round and I am pleased with the revisions made by the authors. I believe this version addressed most of the questions raised earlier and should be ready for publication in *eLife*.

---

## [Author Response]

*Reviewer #1:*

*The last 15 years have seen a marked increase in our knowledge of presynaptic homeostatic plasticity (PHP), largely driven by studies from the Davis lab at the fly NMJ. However, the molecular underpinnings of these events are incompletely known. In this manuscript, Genc and colleagues identify MCTP, a C2-domain containing transmembrane protein, as a new player involved in PHP. It is necessary for both acute and chronic invocations of PHP as well as coordination of PHP over a wide range of calcium concentrations. MCTP localizes broadly throughout nerves, and overlaps somewhat with KDEL, suggesting that it localizes (though not exclusively) to the ER. Beyond its involvement in PHP, MCTP also plays a role in baseline neurotransmission, allowing access to a pool of weakly coupled, EGTA-sensitive vesicles, which may also factor into PHP, though the link is somewhat murky. Perturbation of MCTP does not affect morphology or vesicle parameters at the NMJ. Finally, the authors demonstrate that all three C2 domains of the protein are necessary for PHP, suggesting that the calcium-binding activity of MCTP is what drives its role in PHP.*

Regarding the general significance of our findings: We would like to emphasize that our paper defines the function of a new calcium-binding protein that affects presynaptic neurotransmitter release and homeostatic plasticity. We consider it even more striking that this is an ER-localized protein. We draw attention to the recent work on presynaptic ER calcium signaling in neurons, including recent work publish in Neuron (Juan-Sanz et al., Neuron 2017), now cited in our study.

What distinguishes our work from the existing literature is that we not only identify a novel presynaptic calcium sensor (something remarkable given the long history of the field of synaptic biology), we also place this protein within the presynaptic ER. Thus, we define a new signaling function of the presynaptic ER. Finally, we demonstrate that this protein is a component of the regulatory mechanisms responsible for presynaptic homeostatic plasticity and baseline neurotransmitter release under normal physiological conditions. We would like to emphasize that recent work by Juan-Sanz et al., (Neuron2017) demonstrated that depletion of ER calcium causes a suppression of neurotransmitter release in a Stim1-dependent manner. But, they show that, under normal conditions, *Stim1* knockdown had no effect on baseline release. Our work defines MCTP as essential controller of release and plasticity under normal physiological conditions.

*The experiments are well done, the physiology well controlled, and the data (largely, see below) very strong. There is some question, however, as to how the roles of MCTP regarding PHP and the weakly coupled vesicle population are connected and how separable are the functions of MCTP in baseline transmission and PHP. As such, the mechanistic insight offered by this new player into how PHP functions is limited.*

As stated in our Abstract and Discussion sections, we provide evidence that MCTP has separableactivities to stabilize baseline transmission, short-term release dynamics and PHP. The evidence that MCTP has separable effects on PHP and baseline release is as follows:

a) At 1.5mM extracellular calcium, baseline release is significantly impaired but PHP is fully expressed. Under this condition, the effects are separable.

b) PHP fails in *mctp* mutants as extracellular calcium concentrations are decreased, but baseline release is impaired throughout the calcium range.

c)EGTA-AM normalizes baseline release in wild type and *mctp*, consistent with the conclusion that MCTP is responsible for a weakly coupled vesicle pool. But, EGTA-AM does not block PHP. Again, the effects are separable.

d) Mutations in the three C2 domains of MCTP fail to rescue PHP, but have no effect on baseline transmission compared to the wild type MCTP transgene. Again, the effects are separable.

Thus, it is a conservative conclusion to suggest that MCTP has ‘separable’ effects on baseline release and PHP.

We would also like to emphasize the extent of the information that we provide. We provide strong evidence for how baseline neurotransmission is disrupted (inclusive of EM, calcium imaging, pharmacology and numerous electrophysiological experiments). We also provide strong evidence that MCTP is a novel, presynaptic calcium binding protein necessary for PHP (inclusive a structure function analysis of all three MCTP C2 domains).

Finally, with regard to the intersection of baseline neurotransmitter release and PHP, there are a number of additional points worth making. First, mutations that affect baseline release do notpredict a parallel disruption of PHP. This has been repeatedly demonstrated in the results of our genetic screens and published work (Frank et al., 2006; Frank et al., 2009; Dickman and Davis, 2009; Younger et al., 2013; Goold et al., 2007). Conversely, PHP can be blocked by mutations that have no effect on baseline transmission (Frank et al., 2009; Younger et al., 2013; Wang et al., 2014). Baseline release and PHP are genetically separable processes.Indeed, PHP seems to be a regulatory process that is layered on top of the presynaptic fusion apparatus (Davis and Muller, 2015). There are points of convergence between PHP and the release mechanism including a role for the auxiliary calcium channel subunit (Wang et al., 2016) and the presynaptic cytomatrix (Muller et al., 2012; Muller et al., 2014).

*The feedback system, as suggested by the authors, is a tantalizing prospect, though more insight into that mechanism would be necessary to elevate this paper to eLife. This is an excellent analysis of a new PHP player that would be better suited for a specialty journal.*

We have done our best to consider the implications of our data and have come up with a plausible model that this reviewer considers “tantalizing”. We hope that our model will spark similar interest in other laboratories, in other systems, thereby extending our existing work and ideas. Models are, by definition, unproven ideas. No one will ever put forward a model in a paper if the reviewers request that the model be proven. Models are intended to stimulate speculation and new ideas!

With respect to elevating our work to the level of *eLife*, we refer the reviewer to a new paragraph that we have added to the Discussion section, referring to recent literature. The work that we cite de Juan-Sanz et al. 2017, recently appeared in Neuron. Because our data define a function for MCTP in the control of baseline release and plasticity under physiological conditions, I argue that our data are at least as relevant.

“A recent paper has underscored the role of axonal ER in the regulation of presynaptic release (de Juan-Sanz et al. 2017). In this work, depletion of ER calcium strongly suppresses presynaptic release in a STIM1-dependent manner (de Juan-Sanz et al. 2017). However, STIM-1 knockdown was shown to have no effect on baseline transmission under normal physiological conditions (de Juan-Sanz et al. 2017). By contrast, our work specifies a role for ER-localized MCTP as a critical regulator of baseline release, short-term plasticity and PHP, under physiological conditions.”

Finally, as we now state in our Discussion, the effects of MCTP knockdown are as dramatic as that observed following the knockdown of other important presynaptic proteins such as Rab3 Interacting Molecule (RIM). As such, the phenotypes of *mctp* mutants are rather remarkable for their effects on baseline transmission, downstream of presynaptic calcium influx.

*Specific Points:*

*I am, however, concerned about the conclusions regarding MCTP overexpression. Due to the widespread localization, it is likely to partially colocalize with many markers. So co-staining with Syt, Brp, and HRP is only mildly informative. The KDEL staining is convincing in the segmental nerves (though again, overexpressed), but doesn't match informatively at the NMJ. Thus, I wonder if overexpression causes mislocalization. I would recommend qPCR to compare overexpression levels with those of endogenous. If they are vastly different, identify expression conditions that most closely resemble WT. That would allow for more confidence regarding the localization. Further, commercial KDEL antibodies exist as well as monoclonals to Drosophila Golgi and ER proteins (Riedel et al., 2016, Biology Open) that might be helpful to avoid concomitant KDEL overexpression.*

We appreciate the concern of this reviewer. We have added new experiments and data to address these concerns.

First, we went to great lengths to define, more precisely, the extent of MCTP co- localization with markers of the *Drosophila* ER. This was achieved using new HDEL- GFP transgenes, as suggested by reviewer 3. We also revised our protocols to image HDEL-GFP live, thereby preserving the integrity of the ER morphology. With light fixation and post-fixation processing we are able to precisely correlate MCTP-myc localization with the morphology of HDEL-GFP. Indeed, we think that our data are vastly improved (see new Figure 4). We thank the reviewers for prompting us to do this work. We now show that MCTP-GFP shows very strong co-localization with ER tubules within the presynaptic terminal (see New Figure 4). This complements strong localization in axons and cell bodies of an identified motoneuron.

Second, we moved to heterologous insect cells (S2 cells) for cellular work and an additional structure function analysis of the MCTP protein. These experiments were guided by work on the extended synaptotagmins, done by the DeCamilli laboratory in heterologous cells (Idevall-Hagren et al.). We demonstrate that the transmembrane spanning regions of MCTP are necessary and sufficient for ER-localization of the MCTP protein. The transmembrane region is sufficient to localize GFP to the ER in S2 cells. Conversely, if the transmembrane domains are deleted, the truncated GFP-tagged MCTP protein is no longer retained in the ER. Thus, just like the ER retention signal HDEL, that is a consensus ER retention motif, the transmembrane domains of MCTP function as an ER localization signal.

While there are always caveats to protein overexpression, we would like to point out that this approach remains a powerful tool in modern cell biology. Synaptic vesicle proteins localize to synaptic vesicles when overexpressed. The Rab proteins identify unique endosomal membranes when overexpressed.

When overexpressed, we do not find MCTP appearing on synaptic vesicle membranes (Figure 4) or on the neuronal plasma membrane (Figure 3 and Figure 4), an observation that is now confirmed by work in S2 cells (Figure 3). There is close apposition of these markers in some cases. This is expected given the pervasive tubular architecture of the presynaptic ER. But, there is no evidence of pervasive ‘leak’ of the overexpressed protein onto other membrane systems. Finally, we now present evidence that the transmembrane domains of MCTP are necessary and sufficient for ER protein localization, just as HDEL and KDEL sequences retain GFP in the ER. As this reviewer points out, HDEL-GFP is an excellent ER marker and the same can be said for MCTP based upon our existing and new data. Finally, we demonstrate that overexpression of MCTP rescues a null mutation and does not create a neomorphic phenotype. So, there is no functional evidence for an overexpression artifact.

*In the mctp mutants, the AP waveform is unchanged, but is calcium influx (the currents themselves) is affected? It seems important to know, given the assertions made later on regarding slow calcium buffering with the EGTA experiments.*

The reviewer missed data that we presented in our original submission. We did measure presynaptic calcium transients, in response to both single AP stimulation and short trains of AP stimulation (Figure 6 was dedicated to these data). There is no change in presynaptic calcium influx. This was our stated evidence that MCTP must act downstream of presynaptic calcium influx during baseline neurotransmission and PHP. This was a major point of the original submission and remains so in the revised version.

*I'm particularly intrigued by the change from short-term depression to facilitation in mctp in Figure 5. The same change doesn't appear in Figure 7. Rather, it looks like the level is maintained: is the change only present at certain frequencies? Calcium concentrations? The differences could be made clearer. Or perhaps a more representative trace used?*

This reviewer seems to have missed the quantification of our data that we presented in Figure 5. In this figure, we specifically quantified short-term release dynamics across a full range of extracellular calcium concentrations. The confusion seems to arise because this reviewer is comparing example traces in Figure 5 and Figure 7, and these traces were recorded at different extracellular calcium concentrations (as indicated). The data in Figure 5 were recorded at 1.0 mM and the data in Figure 7 were recorded in 1.5mM Ca^2+^. These sample traces are representative of two very different experiments. Most importantly, the representative traces are precisely representative of the quantified data in Figure 5, calculating short-term release dynamics across a range of extracellular calcium concentrations. The specific calcium concentrations have been made more explicit so as to avoid confusion by other readers.

*Regarding the interpretation of MCTP function, the presence of PHP at high extracellular calcium levels in mctp is interesting. In this case, the calcium driving force is greater, likely overloading normal buffering, or at least slowing it. This would alter the average extra / intra calcium ratio. Is it possible that MCTP is, instead of detecting extracellular calcium, detecting influx, clearance, or extrusion rates from other buffering mechanisms, like PMCA or SERCA? Calcium imaging of transients at high calcium levels may shed some light on the mechanism by which MCTP might work as a sensor. This could further enlighten a mechanistic role.*

In the original submission, we did perform measurements of calcium influx at high (physiological) extracellular calcium (Figure 6). There is no difference in calcium transients in response to single action potential or multiple action potential stimulation. Our calcium measurements show that there is no change in rate of calcium transient decay comparing wild type and MCTP mutants, supporting the conclusion that MCTP does not influence calcium buffering. The question of what calcium MCTP is detecting is a good point, but we can only speculate. What seems clear is that MCTP does not act to alter presynaptic calcium influx.

In a conservative statement in the Results section we write, “We propose that MCTP normally functions to ensure that PHP is robustly and accurately expressed under variable [Ca^2+^]e conditions. To do so, MCTP must have a profound ability to counter changes in extracellular calcium and ensure robust doubling of presynaptic release in the presence of PhTx.” Later, we speculate the following: “Since PHP requires calcium-coordinating residues in all three MCTP C2 domains, we propose that MCTP functions as a presynaptic calcium sensor and a source of calcium-feedback signaling to ensure robust expression of PHP.” We think that we have been careful with the language that we use. Our conclusions stick closely to the data and speculation is referred to as a ‘proposal’ or a ‘model’.

*Reviewer #2:*

*This is an interesting and original study on identification of a presynaptic molecule that plays a key role in homeostatic synaptic plasticity. The author identified an ER- multi-domain Ca sensor from a large scale genetic screen and presented strong electrophysiological evidence that MCTP acts downstream of Ca influx to facilitate the expression of presynaptic compensatory increase in quantal release upon a reduction in postsynaptic glutamate receptors. This work extends previous findings from the Davis lab and appears to add a new presynaptic molecule to the network of proteins involved in homeostatic regulation. What makes this report more interesting is that MCTP is a synaptotagmin-like Ca sensor and localized in presynaptic termini, close to the active zones, to regulate presynaptic release. The data presented in this manuscript are solid and the interpretations are reasonable.*

*1) One of my suggestions is to move Figure 10 to Figure 2 so that the authors can show the structure-function in the context with presynaptic rescue. This is minor.*

We have played around with the order of the figures. We think that there are advantages and disadvantages to the flow of the text. We would prefer, at the end of the day, to leave the structure function analysis to the end.

*2) The facilitation experiment comparing EPSP4 /EPSP1 ratio in Figure 5 is consistent with the observation that the basal release at the mctp mutant NMJs is reduced. A better comparison would be to adjust the extracellular [Ca] such that the initial EPSC is similar for both the wild type and mctp mutant.*

This is a good point. In fact, the data were actually presented in Figure 5. We examined short-term release dynamics over a range of extracellular calcium concentrations as well as baseline release as a function of extracellular calcium.

However, we never explicitly made the comparison that the reviewer suggests. We do so now in the text, drawing attention to specific comparisons in Figure 5.

We now state in the text: “We then compare short-term release dynamics for wild type and mctp under [Ca^2+^]e where baseline release is equivalent. For example, comparing wild type at 0.5mM [Ca^2+^]e with mctp at 0.7mM [Ca^2+^]e it is apparent that baseline EPSCs are equivalent (p>0.05) and so are short-term release dynamics (p>0.05).

These data suggest that the change in short-term release dynamics could be due to a decrease in presynaptic release probability in the mctp mutant compared to wild type.”

*3) The EGTA experiment is intriguing, and I wonder how does this compare with a fast Ca chelator such as BAPTA-AM?*

We thank the reviewer for this comment. BAPTA-AM blocks neurotransmission, consistent with results at other synapses.

*Reviewer #3:*

*In this study, Genc and colleagues report the isolation of a putative smooth ER protein, mctp, which when mutated blocks presynaptic homeostatic plasticity. They postulate that mctp is associated with the smooth ER, which has been previously shown to span the entire neuron, from cell body to synaptic terminals to dendrites. They show that the protein is in close proximity to the plasma membrane and sometimes apposed to active zones. In addition, they provide evidence, including mutations in the C2 domains and alterations in calcium levels, that mctp is likely to function downstream of calcium influx into the cell. They propose that mctp is a calcium sensor that serves to stabilize base transmission and homeostatic plasticity as a function of postsynaptic excitability. These observations are significant as we still know little about the mechanisms of homeostatic plasticity.*

*My main concerns were two-fold, and while they do not detract from most of the main findings in the paper, they do introduce an element of over-interpretation and they should be corrected.*

*1) As has been previously reported, chemical fixation destroys the integrity of the ER in this system (Summerville et al. 2016). Therefore all statements regarding the localization of the mctp signal near the plasma membrane of the terminals, or the localization at ER regions apposed to active zones is in question. Reagents to look at the ER in unfixed preparations are available.*

We are grateful to this reviewer for prompting us to explore this issue in greater detail. We worked hard to find conditions that would preserve the presynaptic ER and we think that we have been quite successful (see new Figure 4). We have imaged HDEL-GFP in a live preparation, followed by mild fixation/antibody labeling and then re-imaging of the same synapses for MCTP localization. The ER tubules in the presynaptic terminal are obvious in the live imaging data, and much better preserved in our new data. As a consequence, the co-localization of MCTP and HDEL is striking. This represents a major improvement to our study and, again, we thank the reviewer for prompting us to work hard on this result (it did take considerable time, with extensive repetitions of trial and error).

With respect to the ER localization at or near active zones: Our new data from both live images and fixed synapses show that the ER is localized near the plasma membrane in tubules that extend throughout the NMJ. Given that active zones are distributed evenly throughout the nerve terminal (at ~400nm centers), it evident that the ER will come in close apposition to many active zones. The image that we show in new Figure 4 also highlights the fact that the ER is not, necessarily, present at active zones. The inset shows that half of a bouton lacks ER membranes, whereas active zones will be distributed throughout all synaptic boutons.

*2) Overexpression of mctp is used to determine the localization of the protein, which is not acceptable, especially given the availability of tools to look at this without overexpression or antibodies (e.g., introduction of a tag into the genomic locus via CRISPR). The arguments presented to argue for overexpression of mctp reflecting the endogenous localization of the protein are not valid as presented (e.g., lack of aggregation of the label might be meaningless as to the normal localization of the protein; that the tagged protein rescues phenotypes can also be meaningless as to protein localization. For example, sufficient amounts of mctp-flag might be present at appropriate locations to rescue abnormal phenotypes, even if mctp-flag overflows to additional sites; the argument that mctp has similar transmembrane domains as another ER protein is also a very weak argument).*

We appreciate the concern of this reviewer and have worked hard to deal with this issue. We have added new experiments and data to our manuscript. First, we went to great lengths to define, more precisely, the extent of co-localization of MCTP with markers of the *Drosophila* ER. This was achieved using new HDEL transgenes, as suggested by this reviewer and improved protocols to preserve the presynaptic ER. Second, we resorted to S2 cells for cellular work and an additional structure function analysis of the MCTP proteins. We now show that MCTP-GFP shows very strong co-localization with ER tubules within the presynaptic terminal (see New Figure 4). This complements strong localization in axons and cell bodies of an identified motoneuron.

We then pursued experiments to demonstrate that the two-transmembrane spanning region of MCTP is necessary and sufficient for ER-retention of the MCTP protein. The transmembrane region is sufficient to localize GFP to the ER in S2 cells. Conversely, if the transmembrane domains are deleted, a GFP-tagged MCTP protein is no longer retained in the ER and becomes cytoplasmic.

3) Another criticism is about the inappropriate interpretation of the statistical analysis. At least in 3 occasions authors point to "a trend (difference in the mean)" even when the statistical analysis says that a difference between the samples cannot be determined at 95% confidence. Further, the authors go on to discuss the significance of the "trend", which is overstretching the data.

To be clear, we define the word ‘trend’ as a difference in a sample mean that fails to reach statistical significance. For example, we previously stated, “There is a trend toward increased quantal content in animals expressing UAS-mctp^C2B^*, but this fails to reach significance.” We considered this to be common usage. However, we agree that there were places in the text that were less explicit. Therefore, we have removed all statements that include the word ‘trend’ from the text, allowing each individual reader to examine our data for themselves, as presented in our figures with the accompanying tests of statistical significance. We agree that this is preferable.

[Editors' note: further revisions were requested prior to acceptance, as described below.]

*Although your rebuttal cited examples that overexpressed proteins reflect the localization of endogenous proteins, this is not guaranteed for a novel protein. We recognize that the revised manuscript contains a lot of new data, and after discussions among reviewers, we agree that the manuscript is acceptable for publication if you clearly point out the limitations of using overexpressed protein to infer the localization of an endogenous protein.*

We have taken this request seriously and have endeavored to write a paragraph that highlights this issue and which cites relevant review articles and primary research studies that deal directly with the issue of defining where proteins reside within a cell. As can be seen from these papers, this is a non trivial issue and one that is worth highlighting. The following paragraph has been added to the text of our Results section.

“We acknowledge that our evidence for ER localization is based on protein overexpression. […] Thus, based on these and related data, we conclude that MCTP is an ER resident protein, distributed on the ER throughout motoneurons.”